# Piezo-catalysis for nondestructive tooth whitening

Yang Wang[1], Xinrong Wen[1], Yanmin Jia[2], Ming Huang[1], Feifei Wang[3], Xuehui Zhang[4], Yunyang Bai[5], Guoliang Yuan [1] & Yaojin Wang [1✉]

The increasing demand for a whiter smile has resulted in an increased popularity for tooth whitening procedures. The most classic hydrogen peroxide-based whitening agents are effective, but can lead to enamel demineralization, gingival irritation, or cytotoxicity. Furthermore, these techniques are excessively time-consuming. Here, we report a nondestructive, harmless and convenient tooth whitening strategy based on a piezo-catalysis effect realized by replacement of abrasives traditionally used in toothpaste with piezoelectric particles. Degradation of organic dyes via piezo-catalysis of BaTiO$_3$ (BTO) nanoparticles was performed under ultrasonic vibration to simulate daily tooth brushing. Teeth stained with black tea, blueberry juice, wine or a combination thereof can be notably whitened by the poled BTO turbid liquid after vibration for 3 h. A similar treatment using unpoled or cubic BTO show negligible tooth whitening effect. Furthermore, the BTO nanoparticle-based piezo-catalysis tooth whitening procedure exhibits remarkably less damage to both enamel and biological cells.

[1] School of Materials Science and Engineering, Nanjing University of Science and Technology, 210094 Nanjing, Jiangsu, China. [2] School of Science, Xi'an University of Posts and Communications, 710121 Xi'an, China. [3] Key Laboratory of Optoelectronic Material and Device, Department of Physics, Shanghai Normal University, 200234 Shanghai, China. [4] Department of Dental Materials, NMPA Key Laboratory for Dental Materials & Dental Medical Devices Testing Center, Peking University School and Hospital of Stomatology, 100081 Beijing, China. [5] Department of Geriatric Dentistry, Peking University School and Hospital of Stomatology, 100081 Beijing, China. ✉email: yjwang@njust.edu.cn

With the development of the aesthetic standard, more people are eager to improve their appearance with a whiter smile. As a result, tooth whitening has developed into one of the fastest growing aesthetic dentistry procedures[1]. However, discolouration and staining of teeth can be easily caused not only by drug ingestion, but also by habitual intake, such as tobacco use, eating dark fruits, drinking certain beverages (e.g. coffee and tea), and consuming certain flavourings (e.g. vinegar)[2].

In order to gain a confident smile, there are several common methods for tooth whitening, such as professional cleaning and polishing, coverage with crowns or veneers, daily toothbrushing with abrasive toothpaste, and bleaching[3]. Both professional procedural cleaning and coverings require grinding or other enamel-cutting steps, which cause irreversible damage. Furthermore these techniques are expensive and time-consuming[4]. The use of toothpaste with water-insoluble abrasives is safe, time- and cost-effective, but decontamination of tooth stains is realized only by the mechanical friction between tooth and abrasive elements, such as aluminium hydroxide, calcium carbonate, and silicas. Thus, abrasive cleaning exhibits limited efficacy and, furthermore, causes slight scratches to the surface of teeth[5]. Tooth bleaching is a chemical method that generally uses high-concentration hydrogen peroxide as an agent[6]. Although this treatment is highly efficient, bleaching with hydrogen peroxide may cause serious side effects, i.e., loss of organic matrix and increase of enamel micro-roughness. Furthermore, it is possible to induce recurrent extrinsic discoloration due to the increased micro-roughness[7,8]. The mechanism of hydrogen peroxide for tooth whitening is the release of unstable, reactive oxygen species during decomposition into water[9], which will attack organic pigment molecules on the surface of teeth, and degrade staining compounds by oxidation[3]. This mechanism suggests that a material with the capability to excite and release reactive oxygen species could be effective as a tooth whitening agent.

Recently, it has been demonstrated that photo-catalysis effect of blue-light-activated $TiO_2$ nano-particles can be used for effective and non-destructive tooth whitening[10]. Compared with the classical $H_2O_2$-based clinical whitening agent, this method is non-destructive to the teeth, but may cause various photo-toxic and photo-allergic reactions, and in turn lead to damage to oral tissue[11,12], because blue light is required as a stimulus to produce reactive oxygen species. Additionally, $TiO_2$-based techniques require customized toothbrush equipped with an inconvenient and expensive blue light source. Therefore, an effective, non-destructive and safe tooth whitening procedure during our daily activities without extra time-consuming is in demand.

The piezoelectric effect, discovered in 1880 by brothers Pierre Curie and Jacques Curie, is the electric charge that accumulates in certain solid materials with non-centrosymmetric structure in response to mechanical stress[13]. It is important to note that piezoelectric materials are ultrasensitive to mechanical vibration, even water flow, muscle movement, and respiration can also induce electrical charges[14–16]. With the ability to convert mechanical stimuli into electrical signals, or vice versa, piezoelectric materials have been widely used for sensors[17,18], transducers[19,20], actuators[21], and energy harvesters for self-powered devices[22,23]. More importantly, emergent materials with large piezoelectric response have been successively discovered[24–26]. Since electrical charges can be induced by mechanical vibration, piezoelectric materials have also been employed as catalysers, termed as piezo-catalysis or mechano-catalysis[27,28]. The effects are similar to photo-catalysis, which is based on photo-induced charges rather than mechanically-induced charges. Some classical piezoelectric materials, such as ZnO[29], BaTiO_3[30,31], and BiFeO_3[32], have been demonstrated as efficient piezo-catalysts.

In view of notable mechanical vibration during our daily tooth brush procedure (Fig. 1a), and nano-sized abrasive particles in toothpaste (Fig. 1b), we propose a nondestructive and safe tooth whitening strategy based on the piezo-catalysis effect, easily realized by replacement of abrasives with piezoelectric nanoparticles. This strategy can be conveniently operated during our daily toothbrushing without extra time-consuming and additional equipment (Fig. 1c).

## Results

**Working principle of piezo-catalysis for tooth whitening**. The working principle of piezo-catalysis for tooth whitening is shown in Fig. 1d–g. On the surface of poled piezoelectric materials, bound charges are balanced by screening charges, thus the material is electrically neutral[33], as shown in Fig. 1d. The amplitude of polarization will be reduced (Fig. 1e) due to compressive stress (i.e., negative strain) based on piezoelectric effect. This in turn can lead to redistribution of charge carriers and release extra screening charges from the surface. As a result, the excess-charges will disperse into solution to become free charges that combine with water molecules to produce reactive species, such as $\bullet OH$ and $\bullet O_2{}^-$[34]. At the maximum applied mechanical stress (Fig. 1f), the bound charges will be minimized. The excess screening charges will continue to release until the material reaches a new electrostatic balance[35]. When the applied force decreases (i.e., positive strain relative to that in Fig. 1f), the polarization will increase, and in turn charges will be adsorbed from the surrounding electrolyte in order to balance the bound charges induced by the piezoelectric effect (Fig. 1g)[36]. Meanwhile, the charges in the electrolyte with opposite polarity to those adsorbed will participate in the redox reaction to produce $\bullet OH$ or $\bullet O_2{}^-$ reactive species again. Thus, a piezoelectric material under periodical stress and in an electrolyte environment will offer continuous charge to produce $\bullet OH$ or $\bullet O_2{}^-$ reactive species for catalysis, which is analogous to the photo-catalysis under light stimuli[37–39].

**Synthesis and structural characterization of BaTiO₃ nanoparticles**. As a demonstration, classical ferroelectric tetragonal $BaTiO_3$ (BTO) nanoparticles are synthesized by the hydrothermal method (see methods section). Figure 2a shows the X-ray diffraction (XRD) pattern of the as-prepared BTO nanoparticles. All diffraction peaks can be well indexed to the perovskite structure and no impurity phases are detected. The inset exhibits an obvious doublet splitting around the $(002)_C$ diffraction peak, indicating that the BTO nanoparticles were perfectly crystallized in tetragonal perovskite structure[40,41]. Figure 2b shows a scanning electron microscopy (SEM) image of BTO nanoparticles, and the size distribution data of the nanoparticles obtained from a $5 \times 5 \mu m$ zone. It can be seen that the BTO nanoparticles have a generally rectangular morphology with an average size of ~130 nm. The crystallographic structure of BTO is further confirmed by transmission electron microscopy (TEM). A typical TEM image of BTO nanoparticles is shown in Fig. 2c, and the scanning TEM image is given in Supplementary Fig. 1a. The TEM results show that the size of local BTO nanoparticles ranges from 100 to 150 nm, which agrees well with the size distribution data obtained from a large scale (see inset of Fig. 1b). Homogeneous dispersion of these elements in the BTO nanoparticles is clearly shown in energy dispersive X-ray spectroscopy (Supplementary Fig. 1b–d). The high resolution TEM image in Fig. 2d indicates a high crystallinity with lattice fringes of (110). The interplanar spacing of 2.828 Å obtained by the selected area electron diffraction pattern in Fig. 2e is consistent with the interplanar distance of (110) planes. Representative amplitude and phase images

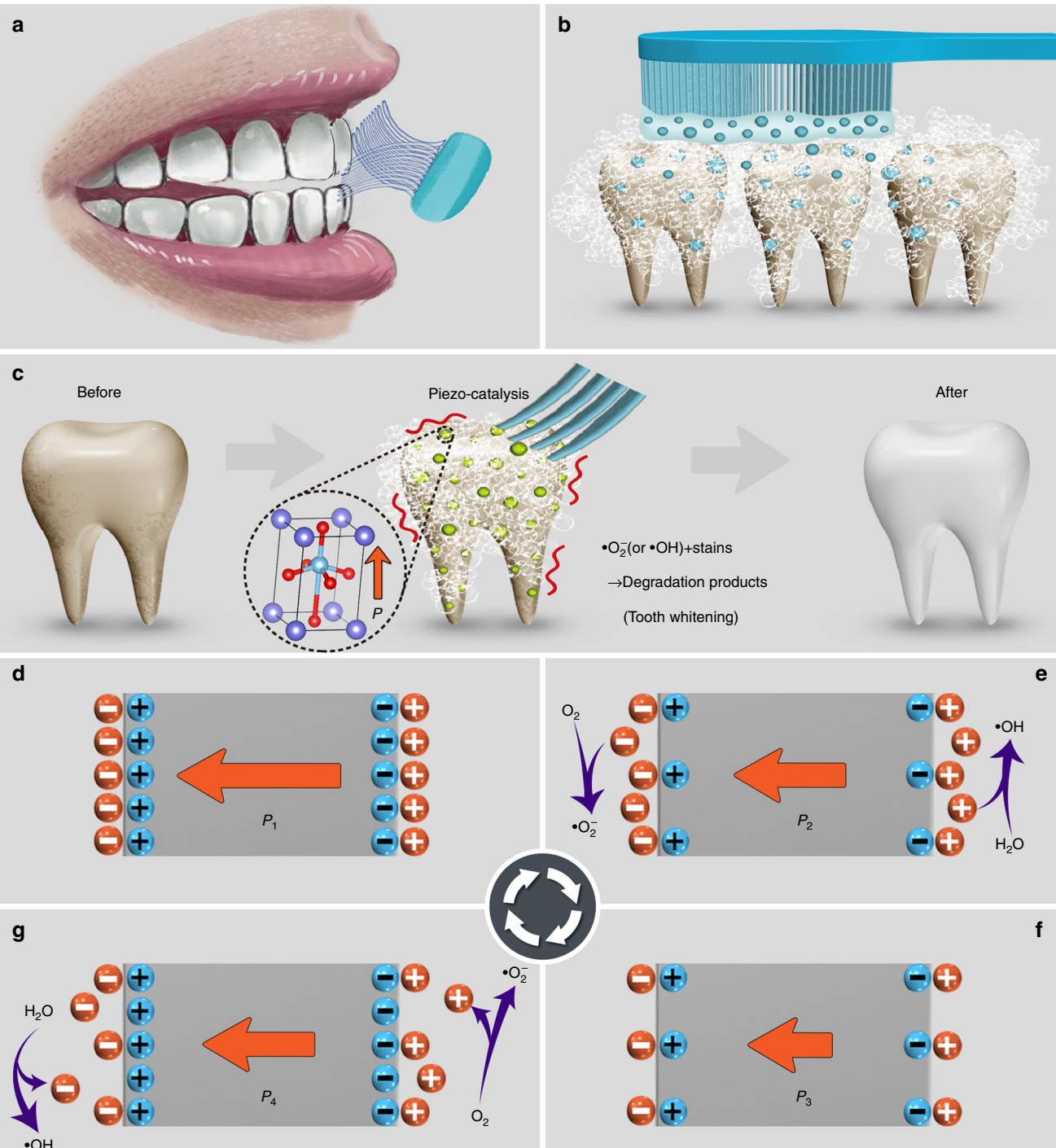

**Fig. 1 Piezo-catalysis effect and working principle for tooth whitening. a** Illustration of mechanical vibration between toothbrush and teeth during daily toothbrushing. **b** The traditional abrasive-based toothpaste method, where tooth whitening is realized by the mechanical friction between teeth and abrasive. **c** The proposed piezo-catalysis effect-based tooth whitening method wherein piezoelectric particles replace traditional abrasive in the toothpaste, to generate reactive oxygen species via piezo-catalysis to bleach tooth stains. The detailed mechanism of piezo-catalysis effect, i.e., **d** the original electrostatic balance state of a poled piezoelectric material; **e** the release of screening charges under compressive stress, which combine with water molecules and then produce reactive species; **f** the modified electrostatic balance state under maximum compressive stress; **g** the adsorption of charges from the surrounding electrolyte under reduced compressive stress. The charges in the electrolyte with opposite polarity to the adsorbed charges will participate in the redox reaction to produce reactive species.

of piezoresponse force microscope are shown in Fig. 2f, g. The amplitude and phase signals of BTO nanoparticles show a clear contrast relative to the silicon substrate, indicating that BTO nanoparticles exhibit robust piezoelectricity and ferroelectricity. The local piezoelectric hysteresis loop of a BTO nanoparticle at

"off" state (i.e., generally regarded as the piezoelectric contribution) is shown in Fig. 2h, while the loop of "On" state is presented in Supplementary Fig. 2. The phase angle shows a 180° change under a 50 V DC bias field, revealing the ferroelectric polarization switching process of BTO nanoparticles. The butterfly-shaped

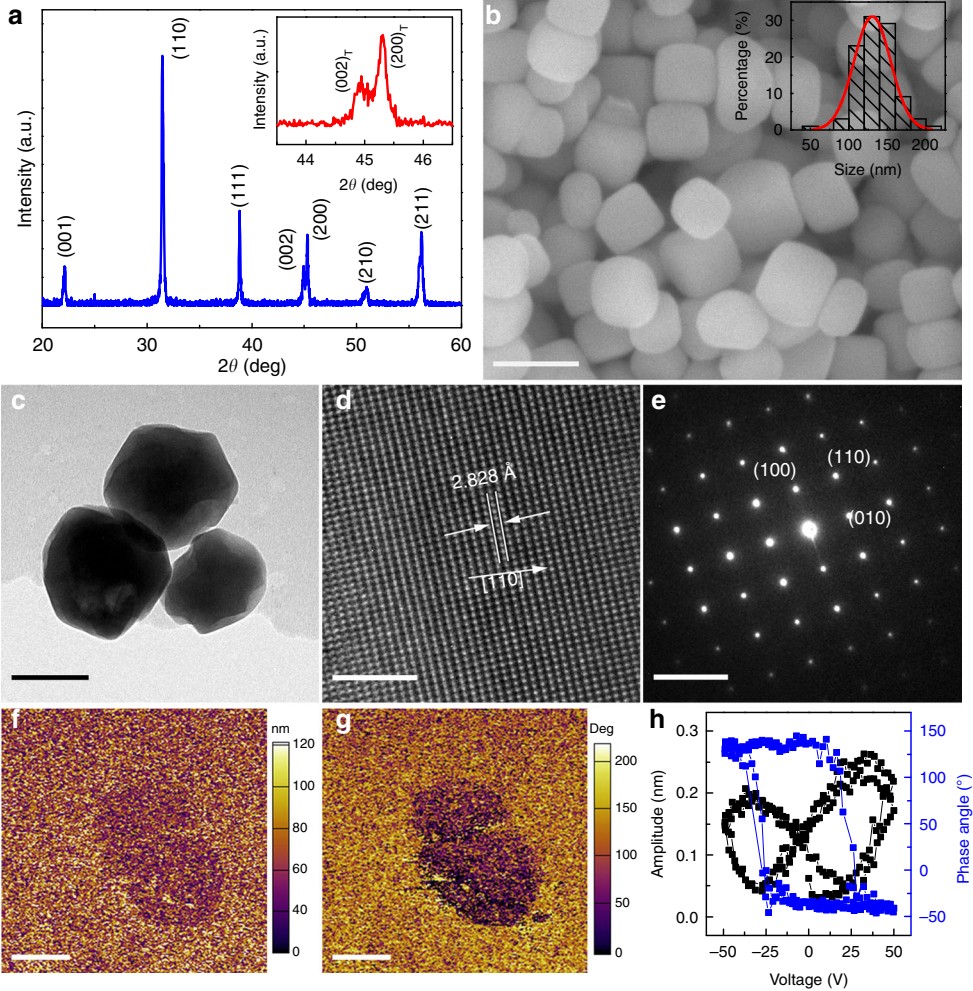

**Fig. 2 Microstructural and morphology characterization. a** X-ray diffraction pattern of the BTO nanoparticles. The inset shows the close-view around
$(002)_C$ diffraction peaks, revealing the BTO nanoparticles have a tetragonal structure. **b** Scanning electron microscope image of BTO nanoparticles, and the
inset shows the particle size distribution of BTO nanoparticles. **c** Transmission electron microscope, **d** high-resolution transmission electron microscope
images and **e** selected area electron diffraction patterns of the BTO nanoparticles. The out-of-plane piezoresponse force microscope images of BTO
nanoparticles for **f** amplitude, **g** phase and **h** piezoelectric hysteresis loop at "off" state. Scale bar: **b** is 200 nm, **c** is 100 nm, **d** is 2 nm, **e** is 5 1/nm,
**f**, **g** is 200 nm.

hysteresis loop further verifies the excellent piezoelectric response
of the BTO nanoparticles.

**Indigo Carmine degradation based on piezo-catalysis**. Since
piezoelectric properties can be significantly improved by electric
poling[42], we first try to pole the as-prepared BTO nanoparticles.
Conventionally, the piezoelectric powder can be poled by a DC
high voltage in a silicone oil bath after being compressed into
pellets and finally mechanically smashing the pellets back into
powders[43]. However, the remnant polarization of piezoelectric
powders may be attenuated due to the multi-domain formation
associated with residual stress release[44,45]. Consequently, we
employed a corona-poling method to directly pole the as-
prepared BTO nanoparticles. As a non-contact method, it
requires a higher electric field to align the polarization of piezo-
electric materials[46]. Supplementary Fig. 3 shows the lab-made
corona-poling setup and working principle. The details can be
found in the Supporting Information Section.

To demonstrate that the piezo-catalysis effect of BTO
nanoparticles can be employed for tooth whitening, the
degradation of Indigo Carmine solution was investigated using

BTO turbid liquid with a concentration of $1\,mg\,mL^{-1}$ under
ultrasonic vibration. Indigo Carmine was selected as a represen-
tative dye since it is a common food colorant used in juice drinks,
carbonated beverages, confected wine and candies, and thus plays
a key role in tooth staining[47]. The ultrasonic vibration was used
to simulate the mechanical stimuli during regular toothbrushing.
The water molecules will be squeezed in the positive phase and
break apart in the negative phase with pressure change, and in
turn cavitation bubbles will be created. It is worth noting that the
pressure on BTO nanoparticles can be up to $10^8$–$10^9$ Pa when a
cavitation bubble ruptures at a critical size (~10 μm)[48].

Then, we evaluate the piezo-catalysis performance of BTO
nanoparticles via the degradation of Indigo Carmine in aqueous
solutions. The UV-Vis absorption spectra of Indigo Carmine
solution at various vibration time for the poled and unpoled BTO
nanoparticles is presented in Fig. 3a, b, respectively. It can be seen
that the Indigo Carmine solution exhibits a maximum absorption
peak around 611 nm, which notably decreases with increased
excitation time for poled BTO nanoparticles, and only slight
changes for unpoled BTO nanoparticles with comparable increase
in mechanical vibration time. The inset of each figure presents a
series of photographs of piezo-catalyzed Indigo Carmine solution

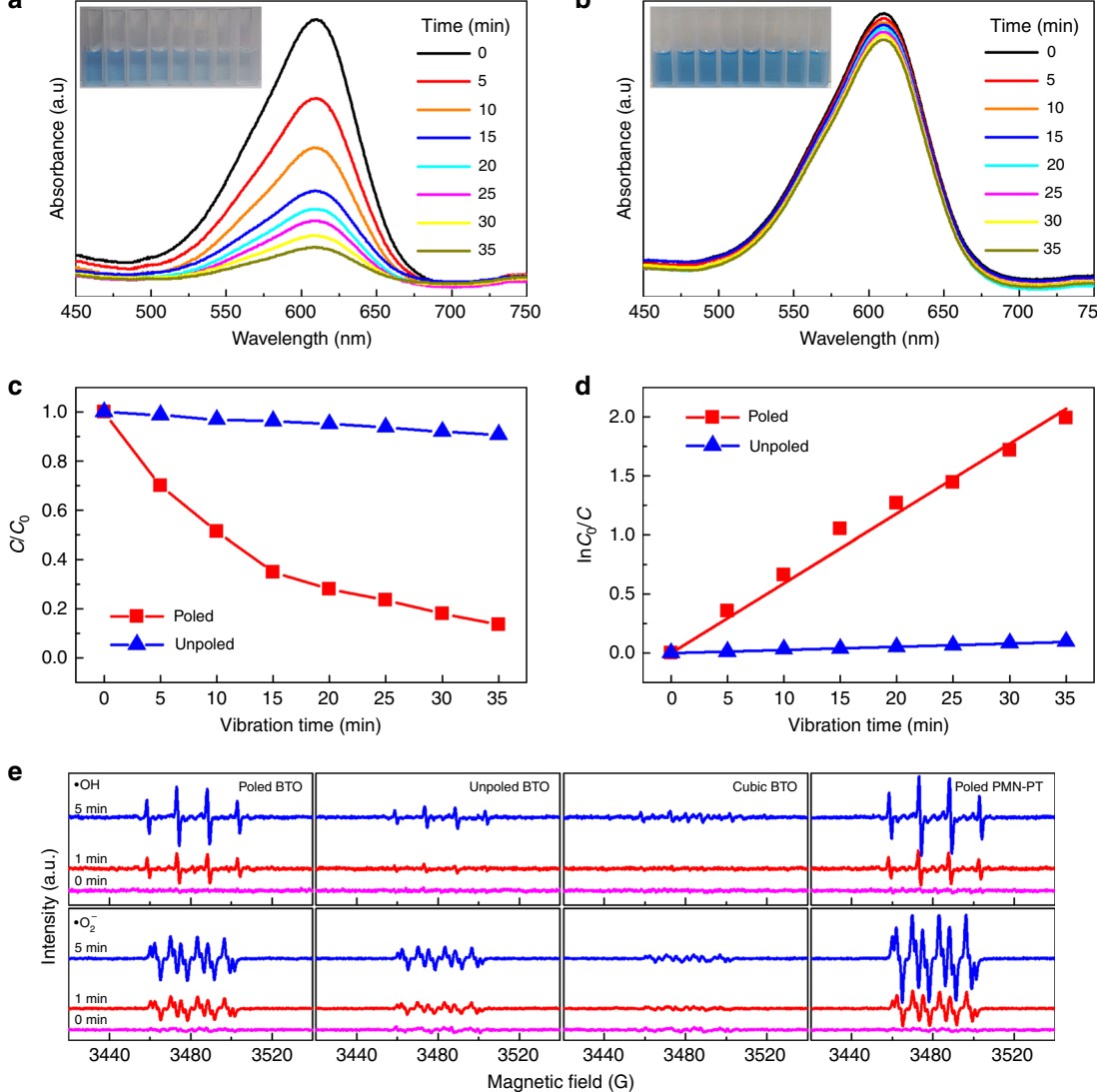

**Fig. 3 Degradation properties of piezo-catalysis.** UV-Vis absorption spectra of Indigo Carmine solutions at various vibration time for the **a** poled and **b** unpoled BTO nanoparticles. The inset in each Figure is a series of photographs of a piezo-catalyzed Indigo Carmine dye solution time progression from left to right. Piezo-catalytic degradation efficiency performance of the poled and unpoled BTO nanoparticles in **c** direct concentration ratio $C/C_0$ and **d** logarithmic relationship of $\ln(C_0/C)$ by fitting with a linear function. **e** Electron paramagnetic resonance spectra (EPR) of radical spin-trapped by 5,5-dimethyl-1-pyrroline N-oxide (DMPO) over different piezo-catalysts in aqueous dispersion (top) and dimethyl sulfoxide (DMSO) dispersion (bottom).

under various vibration time, increasing in time from left to right. The degradation of Indigo Carmine via poled and unpoled BTO nanoparticles can be clearly identified by the naked eye from the color contrast. This comparative experiment unambiguously verifies that the degradation of Indigo Carmine due to the catalysis effect is strongly associated with the piezoelectricity of the nanoparticles.

In order to quantitatively reveal the role of piezoelectricity in vibration-driven degradation of Indigo Carmine, the relative concentration ratio $C/C_0$ of Indigo Carmine for poled and unpoled BTO nanoparticles are plotted as a function of vibration time, where $C$ and $C_0$ are the residual and initial concentrations of the Indigo Carmine solution. As can be seen in Fig. 3c, more than 90% Indigo Carmine was degraded after ultrasonic vibration for 35 min by the poled BTO nano-catalysts. In contrast, the unpoled BTO nanoparticles exhibits negligible degradation effect to the Indigo Carmine solution. The kinetics of the degradation ratio, termed as $\ln(C_0/C)$, was plotted as a function of time (Fig. 3d), and then the rate constant $k$ can be determined by fitting the

experimental data into a pseudo-first-order correlation[49]. It was found that the degradation rate of poled BTO ($k = 0.059\,\text{min}^{-1}$) is about 30 times higher than that of the unpoled BTO ($k = 0.002\,\text{min}^{-1}$).

As Rhodamine B (RhB) has been widely used to identify the catalysis effect, the degradation of RhB was also carried out using both poled and unpoled BTO. Results similar to the Indigo Carmine test were observed (Supplementary Fig. 4). The degradation rate of RhB using poled BTO ($k = 0.448\,\text{h}^{-1}$) is about eight times higher than that of the unpoled BTO ($k = 0.062\,\text{h}^{-1}$). Furthermore, the recyclability of poled BTO for degrading RhB under an ultrasonic vibration was investigated. It was found that the degradation efficiency exhibits no obvious changes after undergoing three recycling processes, indicating the stability of piezo-catalysts for long-term use. The structural stability of nano-sized BTO also serves as evidence that the degradation of organic dye results from the piezo-catalysis effect of BTO, rather than any chemical reaction between BTO and organic dye (see XRD data shown in Supplementary Fig. 5).

To further verify the significance of piezoelectricity in the piezo-catalysis effect, the well-known [001]-oriented Pb(Mg$_{1/3}$Nb$_{2/3}$)-PbTiO$_3$ (PMN-PT) single crystal with composition near the morphotropic phase boundary was used for comparison. Bulk plate specimens with piezoelectric coefficient as high as ~2930 pC N$^{-1}$ (Supplementary Fig. 6a) were broken into pieces ranging in size from hundreds of nanometers to several micrometers by ball-milling (Supplementary Fig. 6b). As expected, the poled PMN-PT powder exhibits an ultrahigh catalysis rate constant of $k \sim 0.036$ min$^{-1}$ or 2.16 h$^{-1}$, and the RhB solution can be completely degraded within 105 min, while the unpoled PMN-PT powder has a much weaker piezo-catalysis effect with a lower rate constant of $k \sim 0.003$ min$^{-1}$ or 0.18 h$^{-1}$ (Supplementary Fig. 6c–f).

The piezoelectricity dependence of degradation of organic dyes can be understood by a series of chemical reactions (Eqs. 1–4). As illustrated in Fig. 1d, the mechanical energy can be transformed into electric charges through the piezoelectricity of nanoparticles (Eq. 1), which combine with water molecules and then produce reactive species (i.e., •OH and •O$_2^-$) (Eqs. 2 and 3). The active species act as the source of the catalytic degradation of the organic dyes (Eq. 4)[50]. It can be seen that the enhanced degradation rate for poled BTO nanoparticles and poled PMN-PT single crystal particles can be attributed to more vibration-induced screen charges (Eq. 1).

$$\text{Piezo} - \text{particles} + \text{Vibration} \rightarrow \text{Piezo} - \text{particles}(q^- + q^+)$$

$$(1)$$

$$q^- + O_2 \rightarrow \bullet O_2^- \qquad (2)$$

$$q^+ + H_2O \rightarrow \bullet OH \qquad (3)$$

$$\bullet OH \text{ or } \bullet O_2^- + \text{organic dye} \rightarrow \text{degradation products} \qquad (4)$$

As the reactive species of •OH and •O$_2^-$ are necessary for piezo-catalysis effect, the generation of •OH and •O$_2^-$ was verified by the electron paramagnetic resonance (EPR) technique using 5,5-dimethyl-1-pyrroline N-oxide (DMPO) as a spin trapper. The signature peaks of both DMPO- •OH (top of Fig. 3e) and DMPO- •O$_2^-$(bottom of Fig. 3e) were detected, and the intensities of the EPR signals increased with vibration time. In addition, the EPR signals are dependent on piezoelectricity (i.e., poled PMN-PT > poled BTO > unpoled BTO > cubic BTO), which is consistent with the results of RhB degradation experiment (Supplementary Figs. 4 and 6).

**Tooth whitening demonstration based on piezo-catalysis**. After the piezo-catalysis effect was verified via Indigo Carmine and RhB degradation, a tooth whitening experiment was carried out using poled BTO nanoparticles. Since habitual intake of beverages and flavorings can result in discoloration and staining of teeth, in-vitro teeth were immersed in liquid mixture of black tea, blueberry juice and wine for 1 week, and then washed using deionized water to remove the remaining dyes. Photographs of the teeth treated with deionized water and BTO nanoparticle turbid liquid vibration conditions at various treatment times were taken of the same tooth with a standard grayscale card as a reference (Fig. 4a). The stained teeth vibrated in a BTO nanoparticle turbid liquid (bottom of Fig. 4a) gradually whitened with increasing vibration time, while the one in deionized water (top of Fig. 4a) showed imperceptible color change under an identical vibration condition. In principle, the tooth was whitened because of a series of chemical reactions which result in the degradation of chromogen, a product of chemical reactions between sugars and amino acids[1]. Briefly, the reactive species (i.e., •OH and •O$_2$) produced by BTO nanoparticles oxidize the multiple conjugated double bond of

large organic molecules that create stains (i.e., chromogen), in turn leading to smaller compounds (i.e, tooth whitening)[51]. To quantitatively characterize the whitening effect, the Commission Internationale De L'Eclairage (CIELab) system was employed to characterize the color change. This system uses three variables to characterize color change: luminance $L$ represents the difference between light ($L = 100$) and dark ($L = 0$), $a$ designates the color values on the red-green axis, and $b$ designates the color values on the blue-yellow axis[52]. The chroma of the tooth at each vibration condition and after various vibration times was characterized, and the values of $L$, $a$, $b$ are given in Fig. 4b–d. It is obvious that the value of $L$ increased, and the values of $a$ and $b$ decreased with vibration time for the BTO nanoparticle turbid liquid condition, while they show much weaker change for the deionized water condition, indicating the teeth have been whitened by the BTO nanoparticle-based piezo-catalysis effect. The color difference of $\Delta E$ was calculated by Eq. 5 to further verify the whitening effect. As shown in Fig. 4e, the value of $\Delta E$ for both vibration conditions increased with vibration time, but is roughly three times larger for the BTO nanoparticle turbid liquid than that for the deionized water. Thus, the quantitative CIELab measurements agree well with the color contrast in Fig. 4a.

$$\Delta E = \sqrt{\Delta L^2 + \Delta a^2 + \Delta b^2} \qquad (5)$$

Additionally, to verify the effect against other staining agents, separate tooth whitening test were performed using various sources of stains, including black tea, red wine, blueberry juice, and vinegar. As shown in Supplementary Figs. 7–10, all the stained teeth were obviously whitened by the poled BTO nanoparticle turbid liquid after 3 h vibration. It is worth noting that the color of roots is much deeper than enamel because of a higher concentration of staining agents on the roots[53]. It can be seen that the color of the roots lightens but the stains still remain after vibration 3 h, while the whole tooth can be completely whitened in the BTO nanoparticle turbid liquid after 10 h of vibration. The time-dependent whitening effect demonstrates the working principle of piezo-catalysis for tooth whitening. Besides the stability of organic dye degradation (Supplementary Fig. 4e) and structure of BTO, the persistent ability of BTO nanoparticles to whiten teeth after prolonged vibration time was also characterized. Three different stained teeth were successively whitened by the same poled BTO within the same vibration time of 10 h, while the tooth whitening effect of the same poled BTO nanoparticle shows no obvious changes (Supplementary Fig. 11).

It is noted that the tooth whitening effect under constant vibration (Supplementary Fig. 10b) was more significant than that under discontinuous vibration (Supplementary Fig. 10a) even for the same time duration (i.e., 3 h). It is possible that the concentration of BTO nanoparticles was gradually decreased in the discontinuous vibration process. Since each tooth was removed from the BTO turbid liquid and washed every hour, nanoparticles deposited on the tooth were washed away, reducing the concentration of BTO nanoparticles on the tooth surface. The different whitening between the two processes apropos indicates that the tooth whitening effect results from the piezo-catalysis of BTO nanoparticles, rather than from a strictly ultrasonic washing procedure. High-performance PMN-PT single crystal powder was also used as catalyzer to demonstrate the tooth whitening procedure. It was found that a more significant tooth whitening can be realized within 1.5 h using PMN-PT single crystals, as compared to BTO (Supplementary Fig. 12a). Again, as with BTO, the whitening effect was pronounced after a continuous vibrations cycle, as opposed to multiple, shorter vibrations (Supplementary Fig. 12b).

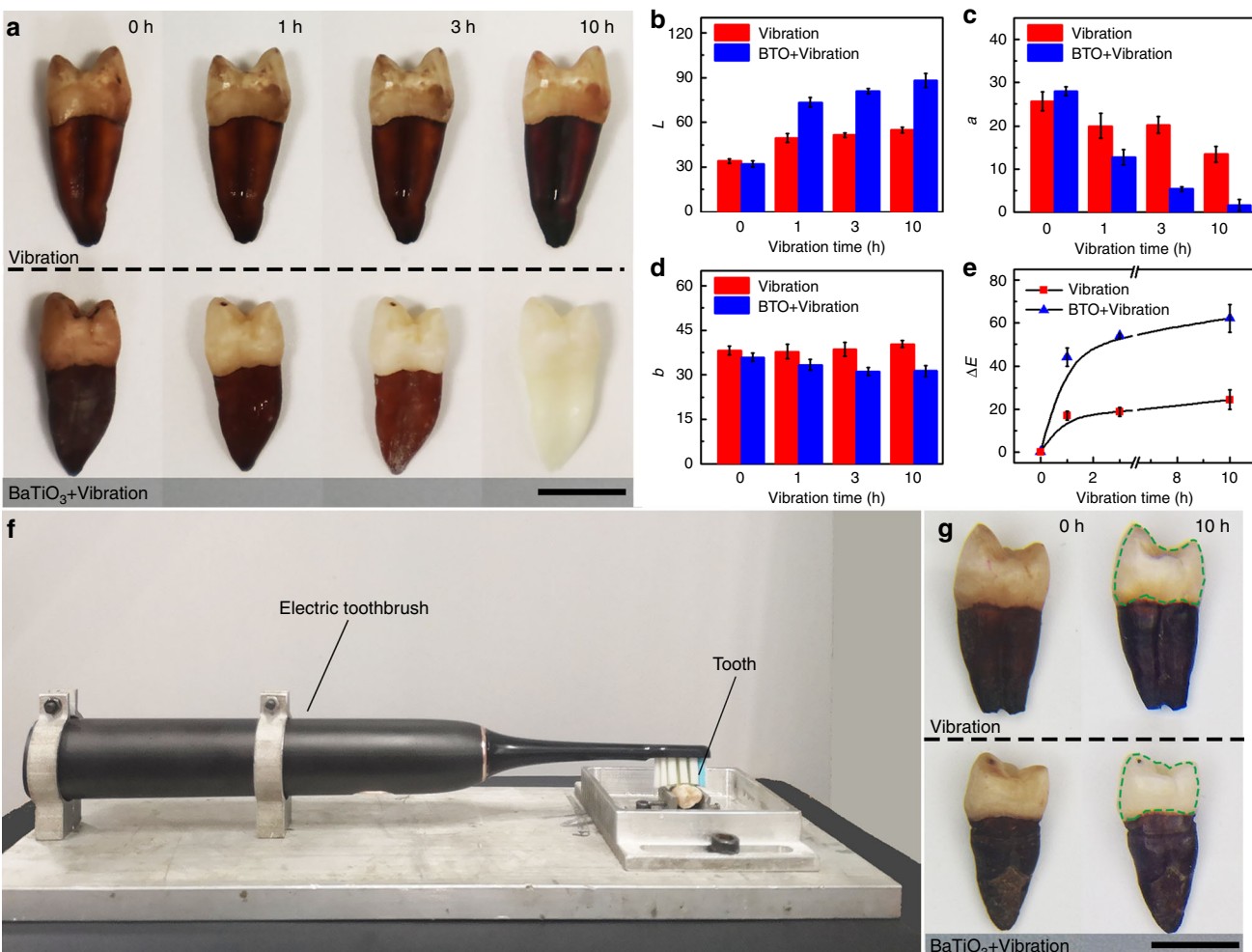

**Fig. 4 Demonstration of tooth whitening based on piezo-catalysis effect. a** Photographs of teeth under treatment of vibration in (top) pure deionized water and (bottom) turbid liquid of BTO nanoparticles for 0, 1, 3, and 10 h, respectively. These photographs are successive images of the same tooth. Variation in **b** luminance $L$, **c** color value of red–green axis $a$, **d** color value of blue-yellow axis $b$ and **e** color difference $\Delta E$ at vibration time of 0, 1, 3 and 10 h. **f** The setup we used to simulate daily teeth cleaning behaviors. **g** Photograph of teeth brushed with pure deionized water (top) and BTO nanoparticles turbid liquid (bottom), respectively. The comparison of brushed zone, marked by circles, reveals that the piezo-catalysis with electric toothbrush was effective to tooth whitening. The difference of $\Delta E$ at each time point was calculated by $t$-test, $p < 0.01$. Scale bars are 1 cm. Error bars = standard deviation ($n = 3$).

A further comparative test was carried out to prove the tooth whitening is caused by the piezo-catalysis, and to rule out the possibility of abrasion by the friction against nanoparticles. BTO nanoparticles with a cubic structure, which has no piezoelectric response, were used as catalysis agents (Supplementary Fig. 13a, b). The BTO nanoparticles obtained by hydrothermal method form with a cubic structure at room temperature, due to the trapped hydroxyl groups in the composition. These cubic particles can be transformed into the tetragonal structure after the hydroxyl groups are removed, for example through annealing at high temperature[54]. The results show that both the solution of RhB and the stained teeth exhibit negligible response with vibration in a cubic nanoparticle turbid liquid (Supplementary Fig. 13c–e). Consequently, these demonstrations unambiguously verify the tooth whitening is due to piezoelectricity-induced piezo-catalysis effect.

In order to further verify the design shown in Fig. 1, an experiment was also carried out using a lab-made electric toothbrush setup (Fig. 4f). The tooth was fixed to a clamp, and an electric toothbrush was used to clean the enamel of stained teeth with deionized water and BTO nanoparticle turbid liquid, as marked by the circles (Fig. 4g). To more realistically simulate

daily toothbrushing, the tooth was vibrated periodically at 2 min intervals for 10 h. The tooth brushed using BTO turbid liquid shows a whitening effect, while in contrast, there was no perceptible change in the color of the one brushed in deionized water alone (Fig. 4g).

The whitening effect realized using a toothbrush is likely not as strong as that of ultrasonic vibration, due to two reasons. First, the tooth was not soaked in the BTO nanoparticle turbid liquid, leading to a decrease of BTO concentration around the enamel. The decrease in BTO will directly reduce the reactive species on the tooth surface. Secondly, the relatively weak vibration energy produced by the electric toothbrush (relative to a laboratory-grade ultrasonic bath) can only stimulate the piezoelectric nanoparticles at the contact surface. Thus there is a smaller amount of reactive species available in the turbid liquid. Although an electric toothbrush shows decreased whitening as compared to the same excitation time of the ultrasonic bath, repetitive, daily toothbrushing results in an extended vibration period, when integrated over multiple months or years. Piezo-catalysis based tooth whitening has been demonstrated using both ultrasonic excitation and with a more traditional, commercial electric

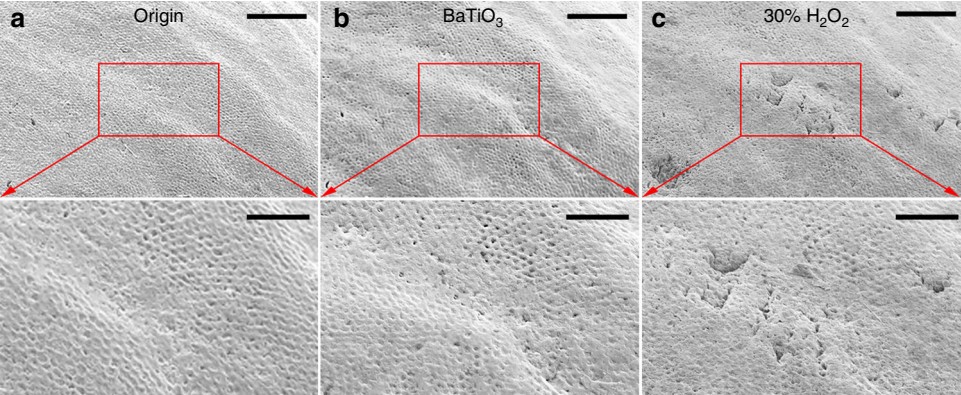

**Fig. 5 Nondestructive characterization.** The scanning electron microscope images of an identical tooth **a** before whitening treatment, **b** after piezo-catalysis whitening in BTO turbid liquid for 3 h, and **c** after further whitening by 30% H₂O₂ for 2 h. The images at the bottom are the enlarged view of the marked identical zone. Scale bars are 100 μm (top) and 50 μm (bottom).

toothbrush. As such, this technique can accelerate the tooth whitening relative to the traditional mechanical friction of abrasive-based tooth pastes alone.

**Nondestructive and harmless nature of piezo-catalytic tooth whitening.** From an application point of view, the effect on enamel and cytotoxicity are essential for a tooth whitening procedure. First, the micro-morphology of a tooth was compared after whitening using a piezo-catalytic technique and a commercial hydrogen peroxide-based tooth bleaching. To form a baseline, the micro-morphology of a tooth was captured using scanning electron microscopy (SEM) (Fig. 5a). The effects of both piezo-catalysis and of bleaching of hydrogen peroxide on enamel were evaluated in the same zone as the baseline reference. Figure 5b presents the SEM image of the tooth after vibration for 3 h in BTO nanoparticle turbid liquid, while another group of SEM images under vibration for 10 h was presented in Supplementary Fig. 14. Please note that the stained tooth can be notably whitened in 3 h and completely whitened within 10 h (Fig. 4a). The SEM image of microscopic morphology shows that the piezo-catalytic tooth whitening is mechanically nondestructive to the tooth enamel. After subsequent treatment using a traditional clinical procedure with a 30% hydrogen peroxide solution for 2 h, damage on the enamel is apparent as corrosion holes and imprints (Fig. 5c). A similar experiment was also carried out using a 3% hydrogen peroxide solution, which is often used clinically to clean the mouth. SEM images show that the degree of enamel damage in the 3% hydrogen peroxide solution is noticeably progressed compared with treatment using BTO (Supplementary Fig. 14). In the 3% hydrogen peroxide solution, the tooth enamel was eroded and extensive damaged, exhibiting pitting and holes. The relative lack of enamel deterioration using BTO nanoparticles further indicates that the reactive species created by piezo-catalysis of BTO nanoparticles are less than that created by 3% H₂O₂ during the tooth whitening process. Furthermore, the concentration of reactive species generated from BTO nanoparticles is not high enough to damage the enamel (Supplementary Fig. 15), whereas the high concentration of reactive species, and violent nature of their creation from H₂O₂ are such that enamel damage is likely[55].

To further explore the non-destructive nature piezo-catalysis, Vickers microhardness was measured during piezo-catalysis based tooth whitening. Three teeth were chosen randomly, and the hardness of the enamel was measured under original, stained, and whitened states, as shown in Supplementary Fig. 16. The hardness is about 300 HV for all the teeth, and the value is stable

across the surface of the tooth. It is obvious that the hardness of the teeth does not vary during either the staining or the whitening process. The stable structural hardness also reveals that tooth whitening results from the piezo-catalysis effect.

To evaluate the biocompatibility of piezo-catalysis, rat arterial smooth muscle cells (A7r5) were evaluated using the MTT method[56]. Fluorescence microscope images for different tissue cultures across three days are presented in Fig. 6a–c. The BTO nanoparticles were confirmed as biocompatible, since the cells showed no changes with time, and there were no obvious differences relative to the control. Figure 6d presents the results of the MTT assay, which clearly shows that BTO nanoparticles have no cytotoxicity to the rat arterial smooth muscle cells, while the cells were almost killed by the 15% hydrogen peroxide solution. This cytotoxicity demonstration shows that BTO nanoparticles are biologically safe and the piezo-catalytic tooth whitening procedure is harmless. Additionally, the possibility of Ba²⁺ leakage during vibration was also examined, and the results show no detectable Ba²⁺ was created after 30 min, which far exceeds typical daily toothbrushing vibration levels (Supplementary Fig. 17).

**Discussion**

In conclusion, we have demonstrated a nondestructive, biocompatible, cost-effective, and time-efficient tooth whitening strategy based on a piezo-catalysis effect arising from piezoelectric nanoparticles. Ferroelectric tetragonal BTO nanoparticles with an average size of ~130 nm were synthesized as a catalyzer by a hydrothermal method. In order to demonstrate the piezo-catalysis effect, the degradation of both Indigo Carmine and RhB were examined by exposure to poled BTO turbid liquid under ultrasonic vibration. A comparative experiment was carried out using unpoled tetragonal BTO, cubic BTO nanoparticles, and high-performance PMN-PT single crystal particles. The degradation rate of organic dyes using these catalyzers verified the catalysis effect was a result of the piezoelectricity. Furthermore, teeth stained with different agents can be notably whitened with a poled BTO turbid liquid after vibrating for 3 h, and it were completely whitened after 10 h, regardless of staining agent type. Additionally, the tooth whitening system using poled BTO nanoparticles shows excellent chemical, structural and electrical stability. The piezo-catalysis based tooth whitening effect was also demonstrated using a more practical electric toothbrush and BTO turbid liquid. Our results indicate that the piezo-catalysis based tooth whitening via BTO nanoparticles is nondestructive to the tooth enamel and biocompatible without cytotoxicity as

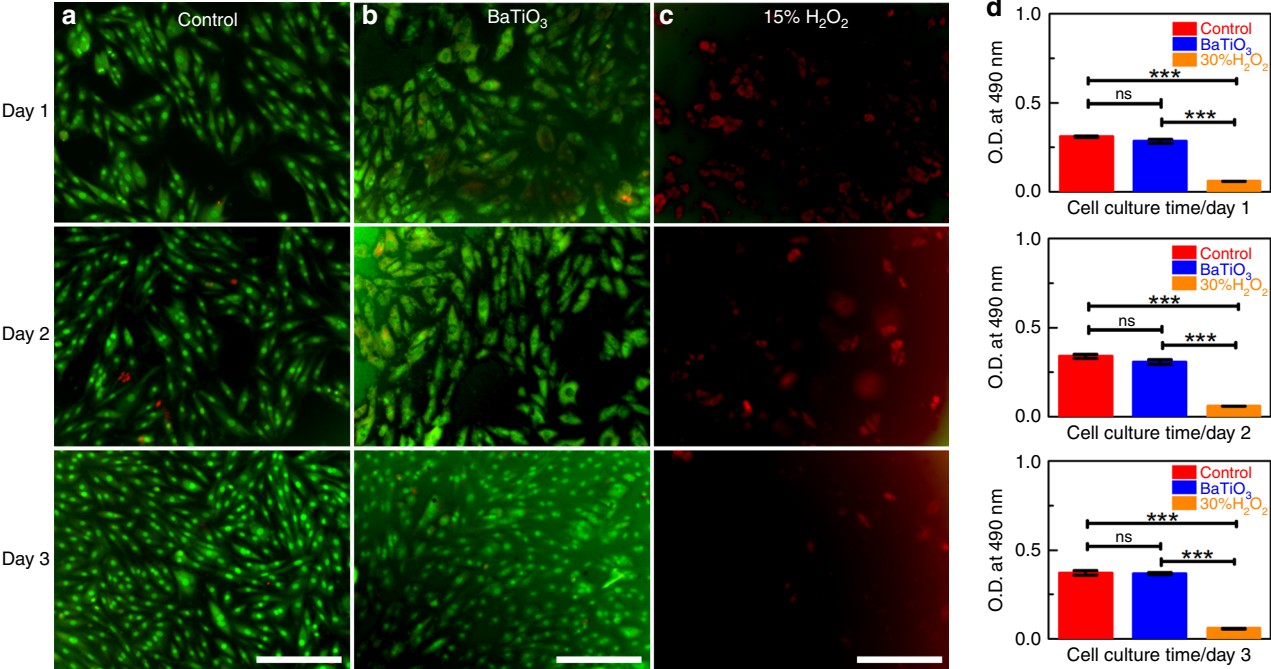

**Fig. 6 Cytotoxicity characterization.** Cell proliferation and morphology with different tissue culture media for 1, 2, and 3 days. AO/EB staining of A7r5 cells grown in **a** pure tissue culture, **b** tissue culture with BTO nanoparticles in concentration of 1 mg mL$^{-1}$, and **c** tissue culture with 15% volume fraction $H_2O_2$. **d** Viability of A7r5 cells in different tissue culture measured by MTT assay. NS = $p > 0.05$, ***$p < 0.01$ ($t$-test at each timepoint). Scale bars are 200 μm. Error bars = standard deviation ($n = 3$).

compared with widely used hydrogen peroxide based clinical whitening agents.

As shown from experiments with electric toothbrushes, the proposed piezo-catalysis based tooth whitening procedure can be effective when incorporated in a daily toothbrushing regime that replaces toothpaste made using traditional, passive abrasives with piezoelectric particles—particularly when used in conjunction with the strong, high frequency excitation present in traditional electric toothbrushes. Unlike existing techniques, piezo-catalysis tooth whitening has the potential to be widely adopted for home use, without requiring significant investment from consumers in either time or resources.

## Methods
**Materials preparation**. The BaTiO$_3$ nanoparticles were synthesized by the hydrothermal method. 3.41 mL Ti(C$_4$H$_9$O)$_4$ was added to 10 mL ethanol. After stirring for 30 min, NH$_3$·H$_2$O was added into the solution dropwise until no more deposits were formed. The suspension of Ti(OH)$_4$ was then transferred into a Teflon-lined autoclave. To maintain a 3:1 molar ratio of Ba$^{2+}$:Ti$^{4+}$, 9.45 g Ba(OH)$_2$·8H$_2$O was dissolved in 20 mL deionized water, after these powders dissolved completely, the solution was mixed with the suspension. Thereafter, the Teflon-lined autoclave was heated at 180 °C for 48 h. Before the reaction, the pH value of the mixture was adjusted to 12 using 6 M KOH solution. The final products were washed in acetic acid, ethanol and deionized water successively several times, and finally dried at 80 °C for 24 h. The cubic BTO nanoparticles used in this work were purchased from Aladdin Co., Shanghai, China.

**Structural characterization**. The phase structure of BaTiO$_3$ nanoparticles were determined by X-ray diffractometry (XRD, Bruker D8) with Cu Kα radiation ($\lambda$ = 1.5406 Å, $2\theta = 20°–60°$). Microstructures were characterized by a field-emission scanning electron microscopy (SEM, ZEISS Merlin), a transmission electron microscopy (TEM) and a field-emission high-resolution transmission electron microscopy (HRTEM, FEI Tecnai G20). The sample was dispersed in ethanol, and a drop of solution was deposited onto a Si (100) substrate to allow the ethanol solvent to evaporate. A piezoresponse force microscopy (PFM, MFP-3D) was used to characterize the piezoelectric performance of the materials. The optical absorption spectra of Indigo Carmine and RhB molecules in the centrifugate was measured to determine the concentration of dye solutions by a Shimadzu UV-3600 UV–VIS–NIR spectrophotometer.

**Piezo-catalytic effect test**. For catalytic tests, BTO nanoparticles were poled using a lab-made corona-poling system. The BTO particles were placed on a copper plate, then poled with a positive corona discharge. In this work, Indigo Carmine and RhB was selected as the target pollutant, 50 mg poled BTO particles and 50 mg unpoled BTO particles were dispersed in 50 mL of 10 mg L$^{-1}$ Indigo Carmine or RhB aqueous solution, respectively. The obtained solutions were stirred for 30 min to reach an adsorption-desorption equilibrium between BTO nanoparticles and Indigo Carmine or RhB molecules. An 80 W, 40 kHz ultrasonic vibration was used to simulate the vibration during daily tooth brushing. In all, 3 mL samples of the suspension were periodically collected and centrifuged to remove the catalysts. Finally, the concentration of Indigo Carmine or RhB remained in clean solution was determined by a UV-Vis spectrophotometer.

**Detection of reactive species**. The reactive species created by the piezocatalyst were detected by the electron paramagnetic resonance (EPR) technique with a Bruker EMX-10/12 spectrometer. First, 10 mg samples were dissolved in 10 mL of deionized water or 10 mL of dimethyl sulfoxide for •OH and •O$_2^-$ detection, respectively. After 15 min of vibration, 200 μL solution was taken out and 20 μL of 5,5-dimethyl-1-pyrroline N-oxide (DMPO) was added into the solution. The reactive species were detected immediately after ultrasound for 0, 1, and 5 min. For the •OH in 3% $H_2O_2$, we used a Fenton reaction to release the reactive species. Briefly, 10 μL of 5 mmol L$^{-1}$ FeSO$_4$ was mixed with 20 μL DMPO, and 180 μL of 3% $H_2O_2$ was added in the mixed solution. Under ultrasonic vibration for 30 s, the •OH created by this classical reaction was detected.

**Tooth whitening experiment**. The extracted teeth (approved by the Ethical Committee of Stomatological School of Nanjing Medical University) we choose were healthy and free of caries. Each tooth was washed with distilled water immediately after removal, and soft tissues attached to periodontal tissues and dental stones was scrapes away. To prevent dehydration, the teeth were soaked in 0.5% Chloramine-T solution (9.0 g sodium chloride and 5.0 g chloramine–trihydrate dissolved in 1000 mL distilled water). Before the tooth whitening experiment, teeth were immersed into black tea, blueberry juice, wine, or a mixture of these liquids for 1 week. After soaking, the teeth were washed using deionized water to remove the remaining staining agents on the surface. Washing continued until the rinse water was clear. Finally the teeth were dried using bibulous paper. The stained teeth were placed in beakers with 50 mL deionized water, 50 mL poled BTO suspension (1 mg mL$^{-1}$) and 50 mL unpoled BTO suspension (1 mg mL$^{-1}$), respectively. After stirring for 30 min, the beakers were ultrasonically vibrated for 10 h. The chroma of the tooth enamel was periodically sampled by computer-aided shade matching (VITA Easyshade). In order to investigate the influence on enamel after whitening, we prepared a colored tooth and observed an area on surface by SEM before whitening. The tooth was

ultrasonically vibrated by in 50 mL poled BTO suspension (1 mg mL$^{-1}$) and 50 mL 30% hydrogen peroxide ($H_2O_2$) successively. The same area on surface was observed by SEM after each whitening experiment.

**Cytotoxicity testing**. Rat arterial smooth muscle cells (A7r5, provide by Shanghai Institute of Cell Biology, Chinese Academy of Sciences) were cultured to logarithmic phase and inoculated in 96-well plates and soaked in complete medium before seeding. BTO nanoparticles and 30%$H_2O_2$ were added, respectively. The BTO turbid liquid had a concentration of 1 mg mL$^{-1}$, and the $H_2O_2$ concentration is 15%. In addition, a control group was set. All the groups were cultured at 37 °C in an incubator under an atmosphere containing 5% $CO_2$. At certain intervals, MTT [3-(4.5-dimethylthiazol-2-yl-2.5-diphenyl) tetrazolium bromide] (5 mg mL$^{-1}$) was added. After 4 h of incubation, the medium was replaced with 150 μL dimethyl sulfoxide to dissolve the precipitates. The absorbance at 490 nm was determined using an ELISA reader (TECAN Infinite 200 Pro, Switzerland). At day 1, 2, and 3 the samples were stained with AO/EB (5 μg mL$^{-1}$ each in PBS). After 3 min of staining, the samples were rinsed with PBS and imaged at the WU module using a fluorescence microscope (Olympus IX81, Japan).

**Reporting summary**. Further information on research design is available in the Nature Research Reporting Summary linked to this article.

## Data availability
The data that support the findings of this study are available from the corresponding author on request.

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

## Acknowledgements

This work was supported by the National Natural Science Foundation of China (51790492, 11874032, 51602156 and 51911530120), the Fundamental Research Funds for the Central Universities (30918012201), and the Opening Project of Key Laboratory of Inorganic function material and device, Chinese Academy of Sciences (KLIFMD-201801).

## Author contributions

Y.J.W. conceived this work and designed the experiments; Y.J.W., Y.W., X.W., M.H., F.W., X. Z., and Y.B. performed the experiments; the data analysis was performed by Y.J.W., Y.W., Y.M.J., and G.L.Y; Y.W. and Y.J.W. wrote the paper. All authors reviewed and commented on the paper.

## Competing interests

The authors declare no competing interests.
