## [Peer Review File · Nature Communications]

Reviewers' Comments:

Reviewer #1:

Remarks to the Author:

The manuscript entitled "Piezo-catalysis for nondestructive tooth whitening" reported a fresh idea about tooth whitening by ferroelectric BaTiO₃ nanoparticles catalysis. Compared with existing tooth whitening method, the BTO-catalysis is nondestructive and harmless. To prove these points, systematic characterization on the efficiency of poled tetragonal BTO nanoparticles and extensive comparisons have been conducted. The obvious contrast between poled and non-poled BTO nanoparticles provide direct evidence for the efficient piezo-catalysis ability. Furthermore, the catalysis efficiency can be greatly enhanced by using PMN-PT with large d₃₃ value. The overall data in the manuscript is abundant and convincing. For the past few years, the piezoelectric field has witnessed the prosperous progress of molecular ferroelectrics with considerable piezoelectricity, some related literatures (such as 10.1126/science.aai8535; 10.1126/science.aav3057) should to be cited for more proper reference review. This manuscript will provide new envision for future application of piezoelectric/ferroelectric materials. I'd like to recommend the publication of this well-written manuscript. By the way, some technical points should be revised or explained.

- 1) The abbreviation RhB should be defined at the first appearance.
- 2) The synthesis of cubic BTO is missing. Is the cubic BTO stable at room temperature? What's the difference between the traditional high temperature (> 393 K) cubic BTO and the room temperature cubic BTO mentioned in the manuscript?

Reviewer #2:

Remarks to the Author:

In this manuscript, the piezo-catalysis is proposed for tooth whitening, the idea is novel and interesting. However, the experiments are technically not sound enough, some concerns need to be settled before further consideration.

(1) What are chemical sources and complex compositions leading to the discoloration and staining of the teeth? More complicated organic and inorganic matter would be involved. A related statistical survey is necessary for different people with diverse living and eating habit. In this regard, the RhB and vinegar removal by piezo-catalysis is NOT enough for demonstrating the tooth whitening effects by piezo-catalysis.

(2) The teeth whitening demonstration is carried out in solution, with the stained teeth and piezo-nanoparticles inside. These experiments may suggest something related. However, those experiments are different from the design shown in Figure 1, and the daily teeth cleaning behaviors. The experiments can be just done as the scheme illustrated, teeth cleaning with traditional abrasive or piezo-nanoparticles in the toothpaste. Or, I suggest that the author try to fix the piezo-nanoparticles on the toothbrush, in this way the piezo-effects would be stronger during tooth brushing.

(3) Since the teeth cleaning by piezo-catalysis also involved the production of ·OH and other highly reactive species, I can not understand the difference in teeth and tissue hurting between piezo-catalysis and H₂O₂ involved treatment. In this regard, more detailed mechanistic studies are required.

Reviewer #3:

Remarks to the Author:

This is an interesting paper that describes a piezoelectric nanoparticle with vibration induced piezo-catalytic properties for reducing chromogens associated with tooth stains. The use of piezoelectric

nanoparticles is not new to dentistry where nano Titanium Oxide (TiO₂) exposed to UV (or blue) light acts as a photocatalytic agent for antimicrobial activity. However the use of BaTiO₃ (BTO) with vibration to cause tooth bleaching is novel.

This study included:

Synthesis and characterization of BTO

Validation that BTO is piezoelectric with vibrational energy

Is not cytotoxic

It can be activated with vibrational energy to produce free radicals that reduce chromogens in the tooth.

The whitening process does not harm the tooth surface.

This study has three significant limitations:

1. The number specimen treated and tested was only 3 teeth and with one stain. It is not possible to generalize the results to other stains and populations.
2. The teeth were dehydrated during sample preparations. This permanently changes the physical and chemical properties of the teeth.
3. The whitening results were not quantified and thus the major outcome of these experiments could not be compared to other whitening technologies or even simple statistical analyses.

Comments:

This paper needs English translation help. There are many errors including mistakes in grammar, spelling, and wrong words used.

The acronym RhB (Rhodamine B) should be defined in the text and a reference about the RhB degradation test provided.

A reference for the MTT method should be provided.

The chemical reaction of the BTO-created free radicles ($\cdot\text{O}_2^-$ and $\cdot\text{OH}$) with the chromogen is not described. The paper by Carey [Tooth Whitening: What we now know. J Evid Based Dent Pract 2014;14:70-6] explains that it is multiple conjugated double bonds within molecules that create the color (stain). H₂O₂ reduces the double bonds thus lessening the color from the chromogen. It is likely that the radicals react with the double bonds to lessen the color.

Specific questions about the assessment of color changes in teeth:

1. The authors should provide some details about the cleaning and storing of extracted teeth. These could alter the outcome of any bleaching study. The teeth need to be stored in an antimicrobial solution and cleaned prior to use. The antimicrobial solution must not alter the collagen structure, or cause the surface to change
2. In the Methods section (Line 358) the authors indicate that they dried the teeth. This should never be done as this irreversibly collapses the collagen structures and the tooth does not react to bleaching in the same way as a natural (never dried) tooth. The dried tooth cannot be rehydrated to be like the natural tooth. The observed whitening of the previously dried teeth in this study may not work the same for natural teeth. Other physical properties are also permanently changed in a tooth when it is dried.
3. In Figure 4 the enamel stains become less but the stains of the roots are not changing as much. The authors should address this observation.
4. There are spectrometric methods available that use the CIELab system to observe enamel color changes and quantify color change. Use of quantification systems would help in describing the effects of the whitening reactions.
5. Does the enamel surface become softer (Knoop Hardness) or change in any way?

For future studies:

- Does the whitening occur in a similar fashion for teeth that have not been dehydrated?

- Does the whitening occur in a similar fashion for teeth stained with other chromogens? I use a combination of black tea, blueberry juice, red wine and fruit punch (Gatorade) and soak the teeth for a week.
- I have seen that dental adhesives and sealants are slowly degraded by peroxide bleaching - does the BTO system alter these polymers?
- I have also seen dental adhesives and sealants take up stains - can the BTO system be incorporated into these materials to prevent stains?

Clifton M. Carey

Reviewer #1:

The manuscript entitled “Piezo-catalysis for nondestructive tooth whitening” reported a fresh idea about tooth whitening by ferroelectric BaTiO₃ nanoparticles catalysis. Compared with existing tooth whitening method, the BTO-catalysis is nondestructive and harmless. To prove these points, systematic characterization on the efficiency of poled tetragonal BTO nanoparticles and extensive comparisons have been conducted. The obvious contrast between poled and non-poled BTO nanoparticles provide direct evidence for the efficient piezo-catalysis ability. Furthermore, the catalysis efficiency can be greatly enhanced by using PMN-PT with large d₃₃ value. The overall data in the manuscript is abundant and convincing. For the past few years, the piezoelectric field has witnessed the prosperous progress of molecular ferroelectrics with considerable piezoelectricity, some related literatures (such as 10.1126/science.aai8535; 10.1126/science.aav3057) should to be cited for more proper reference review. This manuscript will provide new envision for future application of piezoelectric/ferroelectric materials. I’d like to recommend the publication of this well-written manuscript. By the way, some technical points should be revised or explained.

- 1) The abbreviation RhB should be defined at the first appearance.
- 2) The synthesis of cubic BTO is missing. Is the cubic BTO stable at room temperature? What’s the difference between the traditional high temperature (> 393 K) cubic BTO and the room temperature cubic BTO mentioned in the manuscript?

Comment 1: For the past few years, the piezoelectric field has witnessed the prosperous progress of molecular ferroelectrics with considerable piezoelectricity, some related literatures (such as 10.1126/science.aai8535; 10.1126/science.aav3057)

should to be cited for more proper reference review. The abbreviation RhB should be defined at the first appearance.

Response: thanks a lot for your carefully inspect, we have added some related literatures [24] and [25] (i.e., 10.1126/science.aai8535; 10.1126/science.aav3057) in page 3, “More importantly, emergent materials with large piezoelectric response have been successively discovered. [24],[25]”

We have defined “RhB” at the first appearance. In page 10, we changed the “...the degradation of organic dyes (i.e., RhB) was investigated ...” into “...the degradation of organic dyes (i.e., Rhodamine B or RhB) was investigated ...”

Comment 2: The synthesis of cubic BTO is missing. Is the cubic BTO stable at room temperature? What’s the difference between the traditional high temperature (> 393 K) cubic BTO and the room temperature cubic BTO mentioned in the manuscript?

Response: thanks a lot for this crucial comment. The cubic BTO nanoparticles used in this work were purchased from Aladdin Co., Shanghai, China. We determined the crystallographic information by XRD. The cubic BTO at room temperature can exist stably and usually obtained by hydrothermal method, the cubic BTO is produced due to the trapped hydroxyl groups in the composition. If the hydroxyl groups are removed than the cubic structure can be changed to the tetragonal structure [J. Am. Ceram. SOC., 72 [8] 1555-58 (1989)].

We have added the source of cubic BTO in method, and the properties of cubic BTO has been discussed in page 16. “BTO nanoparticles with a cubic structure, which has no piezoelectric response, were used as catalysis (Supplementary Fig. 11 a-b). The BTO nanoparticles obtained by hydrothermal method form with a cubic structure at room temperature due to the trapped hydroxyl groups in the composition, which can be transformed into the tetragonal structure after the hydroxyl groups are removed from the as-prepared materials [52], such as annealing at high temperature.”

The room temperature cubic BTO is due to the trapped hydroxyl groups in the composition, and both of them are paraelectric structure without piezoelectricity.

Reviewer #2:

In this manuscript, the piezo-catalysis is proposed for tooth whitening, the idea is novel and interesting. However, the experiments are technically not sound enough, some concerns need to be settled before further consideration.

(1) What are chemical sources and complex compositions leading to the discoloration and staining of the teeth? More complicated organic and inorganic matter would be involved. A related statistical survey is necessary for different people with diverse living and eating habit. In this regard, the RhB and vinegar removal by piezo-catalysis is NOT enough for demonstrating the tooth whitening effects by piezo-catalysis.

(2) The teeth whitening demonstration is carried out in solution, with the stained teeth and piezo-nanoparticles inside. These experiments may suggest something related. However, those experiments are different from the design shown in Figure 1, and the daily teeth cleaning behaviors. The experiments can be just done as the scheme illustrated, teeth cleaning with traditional abrasive or piezo-nanoparticles in the toothpaste. Or, I suggest that the author try to fix the piezo-nanoparticles on the toothbrush, in this way the piezo-effects would be stronger during tooth brushing.

(3) Since the teeth cleaning by piezo-catalysis also involved the production of $\cdot\text{OH}$ and other highly reactive species, I can not understand the difference in teeth and tissue hurting between piezo-catalysis and H_2O_2 involved treatment. In this regard, more detailed mechanistic studies are required.

Comment 1: What are chemical sources and complex compositions leading to the discoloration and staining of the teeth? More complicated organic and inorganic matter would be involved. A related statistical survey is necessary for different people with diverse living and eating habit. In this regard, the RhB and vinegar removal by piezo-catalysis is NOT enough for demonstrating the tooth whitening effects by piezo-catalysis.

Response: thanks a lot for this valuable comment. Tooth stains consist of compounds that have color or darker shades called chromogens that are accumulated in the tooth (intrinsic) or on the tooth (extrinsic). [J Evid Based Dent Pract, 14 Suppl 70-6 (2014)]. Extrinsic stains usually result from the accumulation of chromogenic substances on the external tooth surface. Extrinsic color changes may occur due to poor oral hygiene, ingestion of chromogenic food and drinks, and tobacco use. These stains are localized mainly in the pellicle and are either generated by the reaction between sugars and amino acids or acquired from the retention of exogenous chromophores in the pellicle. The reaction between sugars and amino acids is called the “Millard reaction” or the “non-enzymatic browning reaction,” and includes chemical rearrangements and reactions between sugars and amino acids. The chemical analysis of stains caused by chromogenic food demonstrates the presence of furfurals and furfuraldehyde derivatives due to this reaction.

In addition, the retention of exogenous chromophores in the pellicle occurs when salivary proteins are selectively attached to the enamel surface through calcium bridges; consequently, a pellicle will form. At the early stage of staining, chromogens interact with the pellicle via hydrogen bridges. Most extrinsic tooth stains can be removed by routine prophylactic procedures. With time, these stains will darken and become more persistent, but they are still highly responsive to bleaching. [Saudi Dent J, 26 [2] 33-46 (2014)]

In page 14, we added a statement about the discoloration and whitening of tooth “In principle, the tooth was whitened because of a series of chemical reactions, which result in the degradation of chromogen, such as product of chemical reaction between sugars and amino acids¹. Briefly, the reactive species (i.e., •OH and •O₂) produced by BTO nanoparticles oxidize the multiple conjugated double bond of large organic molecules that create stains (i.e., chromogen), in turn leading to a smaller compound (i.e, tooth whitening) [49]”

We also used other solutions to stain the tooth, such as vinegar, black tea, red wine and blueberry juice. The teeth were soaked into different staining agents for one

week. The whitening test was taken in the same way as we did in the original manuscript, the results are shown in Supplementary Fig. 6 (black tea), Supplementary Fig. 7 (red wine), Supplementary Fig. 8 (blueberry juice), Supplementary Fig. 9 (vinegar) it can be seen that, all the teeth have been obviously whitened.

We have discussed this test results in our manuscript and all these results have been added in supporting information.

In page 15, we describe the results of teeth with different stains: “Additionally, to avoid the contingency caused by staining agents, separate tooth whitening test were performed using various sources of stains, including black tea, red wine, blueberry juice and vinegar. As shown in Supplementary Fig. 6-9, all the stained teeth were obviously whitened by the poled BTO nanoparticle turbid liquid after 3 h vibration.”

Supplementary Fig. 6. Photographs of teeth stained by black tea under treatment of vibration in (top) pure deionized water and (bottom) turbid liquid of BTO nanoparticles for 0, 1, 3 and 10 hours, respectively. These photographs are successive images of the same tooth. Scale bar is 1 cm.

Supplementary Fig. 7 Photographs of teeth stained by red wine under treatment of vibration in (top) pure deionized water and (bottom) turbid liquid of BTO nanoparticles for 0, 1, 3 and 10 hours, respectively. These photographs are successive images of the same tooth. Scale bar is 1 cm.

Supplementary Fig. 8 Photographs of teeth stained by blueberry juice under

treatment of vibration in (top) pure deionized water and (bottom) turbid liquid of BTO nanoparticles for 0, 1, 3 and 10 hours, respectively. These photographs are successive images of the same tooth. Scale bar is 1 cm.

Supplementary Fig. 9 a Photographs of teeth stained by vinegar under treatment of vibration in (top) pure deionized water and (bottom) turbid liquid of BTO nanoparticles for 0, 1, 2 and 3 hours, respectively. These photographs were taken for an identical tooth every one hours. **b** Photographs for the teeth under consecutive vibration for three hours. The left tooth was whitened in turbid liquid of BTO nanoparticles, and the right teeth was in the pure deionized water as the control. Scale bars are 1 cm.

Comment 2: The teeth whitening demonstration is carried out in solution, with the stained teeth and piezo-nanoparticles inside. These experiments may suggest something related. However, those experiments are different from the design shown in Figure 1, and the daily teeth cleaning behaviors. The experiments can be just done as the scheme illustrated, teeth cleaning with traditional abrasive or piezo-nanoparticles in the toothpaste. Or, I suggest that the author try to fix the piezo-nanoparticles on the toothbrush, in this way the piezo-effects would be stronger during tooth brushing.

Response: thank you for your advice. We have tried to use the electric toothbrush and BTO nanoparticles to realize the tooth whitening by piezo-catalysis. The setup we used can be found in page 13, Figure 4 f-g, the tooth was whitened after 10 hours when BTO turbid liquid used for tooth whitening compared with brushed in water.

Fig. 4 f The setup we used to simulate the daily teeth cleaning behaviors. **g** Photograph of the teeth brushed with pure deionized water (top) and BTO nanoparticles turbid liquid (bottom), respectively. The comparison of brushed zone, marked by circles, reveals that the piezo-catalysis with electric toothbrush was effective to tooth whitening. Scale bars are 1 cm.

This demonstration and the related discussion have been added in page 16

“In order to further verify the design shown in Fig. 1, an experiment was also carried out using a lab-made electric toothbrush setup (Fig. 4f). The tooth was fixed to a clamp, and an electric toothbrush was used to clean the enamel of a stained tooth with deionized water and BTO nanoparticle turbid liquid, as marked by the circles (Fig. 4g). After 10 hours, the tooth brushed using BTO turbid liquid shows a whitening effect, while in contrast, there was no perceptible change in the color of the one brushed in deionized water alone (Fig. 4g).

The whitening effect realized using a tooth brush is likely not as strong as that of ultrasonic vibration, due to two reasons. First, the tooth was not soaked in the BTO nanoparticles turbid liquid, leading to a decrease of BTO concentration around the enamel. The decrease in BTO will directly reduce the reactive species on the tooth surface. Secondly, the relatively weak vibration energy produced by the electric

toothbrush (relative to a laboratory-grade ultrasonic bath) can only stimulate the piezoelectric nanoparticles at the contact surface. Thus there is a smaller amount of reactive species available in solution. Although an electric toothbrush shows decreased whitening as compared to the same excitation time of the ultrasonic bath, repetitive, daily tooth brushing results in an extended vibration period, when integrated over multiple months or years. Piezo-catalysis-based tooth whitening has been demonstrated using both ultrasonic excitation and with a more traditional, commercial electric toothbrush. As such, this technique can accelerate the tooth whitening relative to the traditional mechanical friction of abrasive-based tooth pastes alone.”

Comment 3: Since the teeth cleaning by piezo-catalysis also involved the production of $\cdot\text{OH}$ and other highly reactive species, I cannot understand the difference in teeth and tissue hurting between piezo-catalysis and H_2O_2 involved treatment. In this regard, more detailed mechanistic studies are required.

Response: thanks a lot for this valuable comment. For the H_2O_2 , it has a high concentration with $\cdot\text{OH}$ and other highly reactive species, so the reaction for tooth whitening on the surface of enamel is violently [J Endod, 30 [1] 45-50 (2004)], but for our method, the reactive species created by BTO due to the piezoelectric effect is stepwise and slow. This important difference has been discussed in page 18.

“The reason for this notable contrast is that the creation of reactive species by piezo-catalysis of BTO nanoparticles is stepwise and slow. Furthermore, the concentration of reactive species is not high enough to damage the enamel, while the creation of reactive species on the surface of tooth is violently when H_2O_2 was hired as the whitening agent and the high concentration of reactive species can make the enamel be damaged [53].”

Reviewer #3

This is an interesting paper that describes a piezoelectric nanoparticle with vibration induced piezo-catalytic properties for reducing chromogens associated with tooth stains. The use of piezoelectric nanoparticles is not new to dentistry where nano Titanium Oxide (TiO₂) exposed to UV (or blue) light acts as a photocatalytic agent for antimicrobial activity. However the use of BaTiO₃ (BTO) with vibration to cause tooth bleaching is novel.

This study included:

Synthesis and characterization of BTO

Validation that BTO is piezoelectric with vibrational energy

Is not cytotoxic

It can be activated with vibrational energy to produce free radicals that reduce chromogens in the tooth.

The whitening process does not harm the tooth surface.

This study has three significant limitations:

1. The number specimen treated and tested was only 3 teeth and with one stain. It is not possible to generalize the results to other stains and populations.
2. The teeth were dehydrated during sample preparations. This permanently changes the physical and chemical properties of the teeth.
3. The whitening results were not quantified and thus the major outcome of these experiments could not be compared to other whitening technologies or even simple statistical analyses.

Comments:

This paper needs English translation help. There are many errors including mistakes in grammar, spelling, and wrong words used.

The acronym RhB (Rhodamine B) should be defined in the text and a reference about the RhB degradation test provided.

A reference for the MTT method should be provided.

The chemical reaction of the BTO-created free radicals ($\cdot\text{O}^{2-}$ and $\cdot\text{OH}$) with the chromogen is not described. The paper by Carey [Tooth Whitening: What we now know. J Evid Based Dent Pract 2014;14:70-6] explains that it is multiple conjugated double bonds within molecules that create the color (stain). H_2O_2 reduces the double bonds thus lessening the color from the chromogen. It is likely that the radicals react with the double bonds to lessen the color.

Specific questions about the assessment of color changes in teeth:

1. The authors should provide some details about the cleaning and storing of extracted teeth. These could alter the outcome of any bleaching study. The teeth need to be stored in an antimicrobial solution and cleaned prior to use. The antimicrobial solution must not alter the collagen structure, or cause the surface to change
2. In the Methods section (Line 358) the authors indicate that they dried the teeth. This should never be done as this irreversibly collapses the collagen structures and the tooth does not react to bleaching in the same way as a natural (never dried) tooth. The dried tooth cannot be rehydrated to be like the natural tooth. The observed whitening of the previously dried teeth in this study may not work the same for natural teeth. Other physical properties are also permanently changed in a tooth when it is dried.
3. In Figure 4 the enamel stains become less but the stains of the roots are not changing as much. The authors should address this observation.
4. There are spectrometric methods available that use the CIELab system to observe enamel color changes and quantify color change. Use of quantification systems would help in describing the effects of the whitening reactions.
5. Does the enamel surface become softer (Knoop Hardness) or change in any way?

For future studies:

- Does the whitening occur in a similar fashion for teeth that have not been dehydrated?

- Does the whitening occur in a similar fashion for teeth stained with other chromogens? I use a combination of black tea, blueberry juice, red wine and fruit punch Gatorade) and soak the teeth for a week.
- I have seen that dental adhesives and sealants are slowly degraded by peroxide bleaching - does the BTO system alter these polymers?
- I have also seen dental adhesives and sealants take up stains – can the BTO system be incorporated into these materials to prevent stains?

Limitation 1: The number specimen treated and tested was only 3 teeth and with one stain. It is not possible to generalize the results to other stains and populations.

Response: thanks a lot for this crucial comment, we strongly agreed that it is necessary to do more test with different stains, so more stains have been studied. The black tea, blueberry juice, red wine and the mix liquid of them were used to stain the teeth. For each stain, we did the tooth whitening test using the same method as that in the original manuscript. All the teeth with different stains has been whitened by vibration with poled BTO nanoparticles for 3 hours. All these results have been added in supporting information, see Supplementary Fig. 6 (black tea), Supplementary Fig. 7 (red wine), Supplementary Fig. 8 (blueberry juice) and Supplementary Fig. 9 (vinegar). Presently, we have performed the piezo-catalysis based tooth whitening using more than 13 teeth (2 teeth stained by mix liquid of black tea, blueberry juice and wine; 1 teeth stained by black tea; 1 teeth stained by red wine; 1 stained by blueberry juice, 4 teeth stained by vinegar; 1 tooth for nondestructive test and 3 teeth for microhardness test) in the manuscript and supporting information. Actually, more teeth were tested in our experiment, but not shown in the manuscript and supporting information.

In page 15, we describe the results of tooth with different stains: “Additionally, to avoid the contingency caused by staining agents, separate tooth whitening test were performed using various sources of stains, including black tea, red wine, blueberry

juice and vinegar. As shown in Supplementary Fig. 6-9, all the stained teeth were obviously whitened by the poled BTO nanoparticle turbid liquid after 3 h vibration.”

Supplementary Fig. 6. Photographs of teeth stained by black tea under treatment of vibration in (top) pure deionized water and (bottom) turbid liquid of BTO nanoparticles for 0, 1, 3 and 10 hours, respectively. These photographs are successive images of the same tooth. Scale bar is 1 cm.

Supplementary Fig. 7 Photographs of teeth stained by red wine under treatment of vibration in (top) pure deionized water and (bottom) turbid liquid of BTO nanoparticles for 0, 1, 3 and 10 hours, respectively. These photographs are successive images of the same tooth. Scale bar is 1 cm.

Supplementary Fig. 8 Photographs of teeth stained by blueberry juice under

treatment of vibration in (top) pure deionized water and (bottom) turbid liquid of BTO nanoparticles for 0, 1, 3 and 10 hours, respectively. These photographs are successive images of the same tooth. Scale bar is 1 cm.

Supplementary Fig. 9 a Photographs of teeth stained by vinegar under treatment of vibration in (top) pure deionized water and (bottom) turbid liquid of BTO nanoparticles for 0, 1, 2 and 3 hours, respectively. These photographs were taken for an identical tooth every one hours. **b** Photographs for the teeth under consecutive vibration for three hours. The left tooth was whitened in turbid liquid of BTO nanoparticles, and the right teeth was in the pure deionized water as the control. Scale bars are 1 cm.

Limitation 2: The teeth were dehydrated during sample preparations. This permanently changes the physical and chemical properties of the teeth.

Response: thanks a lot for this valuable comment. The aim of this step is to remove the water adsorbed on the surface of tooth. The temperature is 60°C, but the heating time is very short (< 60 seconds). This dry process is to clean the water drop on the tooth, and then we can take a better photograph for the stained tooth in the revised manuscript. More experiment has been taken under consideration of your valuable concern. In order to avoid the influence of temperature, we used the bibulous paper to

remove the water drops on the tooth. In page 22, we changed “finally dried at 60□” into “Finally the teeth were dried using bibulous paper”.

Limitation 3: The whitening results were not quantified and thus the major outcome of these experiments could not be compared to other whitening technologies or even simple statistical analyses.

Response: thank you for your valuable comment. It is really important to quantify the whitening effects, so we take your advice to use the VITA Easshade CIELab system to observe enamel and quantify color change. The characterization was done with cooperation of Prof. Xuehui Zhang and Dr. Yunyang Bai, so we added two authors (Xuehui Zhang and Yunyang Bai) in the author list. Furthermore, we calibrated the photos using a standard grayscale card.

In page 14: we added a statement “Photographs of the teeth treated with deionized water and BTO nanoparticle turbid liquid vibration conditions at various treatment times were taken of the same tooth with a standard grayscale card as a reference (Fig. 4a)”

Fig. 4 Demonstration of tooth whitening based on piezo-catalysis effect. **a** Photographs of teeth under treatment of vibration in (top) pure deionized water and (bottom) turbid liquid of BTO nanoparticles for 0, 1, 3 and 10 hours, respectively. These photographs are successive images of the same tooth. Variation in **b** luminance L , **c** color value of red-green axis a , **d** color value of blue-yellow axis b and **e** color difference ΔE at vibration time of 0, 1, 3 and 10 hours. **f** The setup we used to simulate the daily teeth cleaning behaviors. **g** Photograph of the teeth brushed with pure deionized water (top) and BTO nanoparticles turbid liquid (bottom), respectively. The comparison of brushed zone, marked by circles, reveals that the piezo-catalysis with electric toothbrush was effective to tooth whitening. Scale bars are 1 cm.

In page 13, we changed the Figure 4, and discussed the quantified whitening results in page 14.

“In order to quantitatively characterize the whitening effect, Commission Internationale De L’Eclairage (CIELab) system was employed to quantify the color change. This system uses three variables: i.e., luminance L represents the difference between light ($L = 100$) and dark ($L = 0$); a and b designate the color values on the red-green axis and blue-yellow axis, respectively⁵⁰. The chroma of the tooth at each vibration condition and various vibration time was characterized, and the values of L , a , b are given in Fig. 4 b-d. It is obvious that the value of L increased, and the values of a and b decreased with vibration time for the BTO nanoparticle turbid liquid condition, while they show much weaker change for the deionized water condition, indicating the teeth has been obviously whitened by the BTO nanoparticle-based piezo-catalysis effect. The color difference of ΔE was calculated by Eq.5 to further verify the whitening effect. As shown in Fig. 4e, the value of ΔE for both vibration condition increased with vibration time, but is roughly 3 times larger for the BTO nanoparticle turbid liquid than that for the deionized water. Thus, the quantitative CIELab measurements agree well with the color contrast in Fig. 4a.

$$\Delta E = \sqrt{\Delta L^2 + \Delta a^2 + \Delta b^2} \quad (5)''$$

The origin Figure 4 was putted into supporting information as Figure S9

Supplementary Fig. 9 a Photographs of teeth stained by vinegar under treatment of vibration in (top) pure deionized water and (bottom) turbid liquid of BTO nanoparticles for 0, 1, 2 and 3 hours, respectively. These photographs were taken for an identical tooth every one hours. **b** Photographs for the teeth under consecutive vibration for three hours. The left tooth was whitened in turbid liquid of BTO nanoparticles, and the right teeth was in the pure deionized water as the control. Scale bars are 1 cm.

Comment 1: This paper needs English translation help. There are many errors including mistakes in grammar, spelling, and wrong words used.

The acronym RhB (Rhodamine B) should be defined in the text and a reference about the RhB degradation test provided.

Response: thanks a lot for your high evaluation to our manuscript. We have invited a native English speaker to polish the manuscript, and we have carefully checked the manuscript and corrected grammatical errors, which are marked in red color in the text.

We have listed some representative changes:

In page 1, “As an exemplary demonstration, ferroelectric tetragonal BaTiO₃ (BTO) nanoparticles with an average size of ~130 nm were synthesized by a hydrothermal method.” “was” was changed to “were”, “the” changed to “a”.

In page 1, “A Rhodamine B degradation test via piezo-catalysis of BaTiO₃ (BTO) nanoparticles was performed under ultrasonic vibration to simulate the experienced during daily tooth brushing.” “RhB” has been defined, and “teeth brush” was corrected as “teeth brushing”.

In page 2, “Both professional procedural cleaning and coverings require grinding or other enamel-cutting steps, which cause irreversible damage. Furthermore these techniques are expensive and time-consuming [4].” This sentence was rewrite in simple language.

In page 4, “As a result, the excess-charges will dispersed into solution to become free charges that combine with water molecules to produce reactive species” We used “dispersed into” to replace “get into”.

In page 4, “Thus, a piezoelectric material under periodical stress and in an electrolyte environment will offer continuous charge to produce •OH or •O₂⁻ reactive species for catalysis” “incessant” was changed to “continuous”.

In page 4, “At the maximum applied mechanical stress (Fig. 1d-iii), the bound charges will be minimized.”

In page 5, “The proposed piezo-catalysis effect-based tooth whitening method wherein piezoelectric particles replace traditional abrasive in the toothpaste” “via” was changed to “wherein”.

In page 9, “The inset in each Figure is a series of photographs of piezo-catalyzed RhB dye solution progressing in time from left to right” We added an annotation for the inset.

In page 10, “As a non-contact method, it requires a higher electric field to align the polarization of piezoelectric materials [45]”

In page 12, “The active species act as the source of the catalytic degradation of RhB” We changed “in source of” to “act as the source of”.

In page 14, “the tooth whitening experiment was carried out **using** poled BTO nanoparticles.” “based on” was changed to “using”.

In page 15, “**Since each tooth was removed from solution and washed every hour,** nanoparticles deposited on the tooth were washed away, reducing the concentration of BTO nanoparticles in the solution.” We rewrite this sentence.

In page 16, “The results show that both the solution of RhB or the stained tooth shows negligible response **with vibration in a cubic nanoparticle solution**” We added information in this sentence.

In page 17, “**To form a baseline, the micro-morphology of a tooth was captured** using scanning electron microscopy (SEM)” We use a simple sentence replaced the origin one.

In page 19, “**To evaluate the biocompatibility, rat arterial smooth muscle cells (A7r5) were evaluated using the MTT method**” We put two sentence together, and it is easier to understand.

In page 20, “In conclusion, we have demonstrated a nondestructive, **biocompatible,** cost-effective, **and time-efficient** tooth whitening strategy based on **a** piezo-catalysis effect **arising from** piezoelectric nanoparticles.” We use “biocompatible” replaced “harmless”, added an advantage of our work “time-efficient”, “of” was replaced by “arising from”.

Please note that all the changes of “mistakes in grammar, spelling, and wrong words” are marked in red, and we did not show all the changes in the response letter.

Furthermore, we also have reorganized the acronyms to ensure that all definitions were given at the first use.

In page 10, “**...the degradation of organic dyes (i.e., Rhodamine B or RhB) was investigated ...**”

Comment 2: A reference for the MTT method should be provided.

Response: thanks for this reminder, a tightly related reference [54] about the MTT method has been added in page 16.

In page 19, “To evaluate the biocompatibility rat arterial smooth muscle cells (A7r5) were evaluated using the MTT method [54].”

Comment 3: The chemical reaction of the BTO-created free radicles ($\cdot\text{O}^{2-}$ and $\cdot\text{OH}$) with the chromogen is not described. The paper by Carey [Tooth Whitening: What we now know. J Evid Based Dent Pract 2014;14:70-6] explains that it is multiple conjugated double bonds within molecules that create the color (stain). H_2O_2 reduces the double bonds thus lessening the color from the chromogen. It is likely that the radicals react with the double bonds to lessen the color.

Response: thanks for your thoughtful question. We have read this article carefully, the chemical reaction during tooth whitening has been described in detail. It is very important for us to classify the mechanism of tooth whitening by BTO piezoelectric effect. The mechanism of tooth whitening has been added in page 14.

“In principle, the tooth was whitened because of a series of chemical reactions, which result in the degradation of chromogen, such as product of chemical reaction between sugars and amino acids [1]. Briefly, the reactive species (i.e., $\cdot\text{OH}$ and $\cdot\text{O}_2$) produced by BTO nanoparticles oxidize the multiple conjugated double bond of large organic molecules that create stains (i.e., chromogen), in turn leading to a smaller compound (i.e, tooth whitening) [49]”

Question 1: The authors should provide some details about the cleaning and storing of extracted teeth. These could alter the outcome of any bleaching study. The teeth need to be stored in an antimicrobial solution and cleaned prior to use. The antimicrobial solution must not alter the collagen structure, or cause the surface to change

Response: thank you for this reminder, the details about storing teeth has been added in page 22.

“The extracted teeth we choose were healthy and free of caries. Each tooth was washed with distilled water immediately after removal, and soft tissues attached to

periodontal tissues and dental stones was scraped away. To prevent dehydration, the teeth were soaked in 0.5% Chloramine-T solution (9.0 g sodium chloride and 5.0 g chloramine-trihydrate dissolved in 1000 ml distilled water).”

Question 2: In the Methods section (Line 358) the authors indicate that they dried the teeth. This should never be done as this irreversibly collapses the collagen structures and the tooth does not react to bleaching in the same way as a natural (never dried) tooth. The dried tooth cannot be rehydrated to be like the natural tooth. The observed whitening of the previously dried teeth in this study may not work the same for natural teeth. Other physical properties are also permanently changed in a tooth when it is dried.

Response: thanks for this critical comment, the aim of this step is to remove the water adsorbed on the surface of tooth. The temperature is 60°C, but the heating time is very short, (< 60 seconds). This dry process is to clean the water drop on the tooth, and then we can take a better photograph for the origin stained tooth in the revised manuscript, more experiment has been taken under consideration of your valuable concern. In order to avoid the influence of temperature, we used the bibulous paper to remove the water drops on the tooth. In page 22, we changed “finally dried at 60°C” into “Finally the teeth were dried using bibulous paper”

Question 3: In Figure 4 the enamel stains become less but the stains of the roots are not changing as much. The authors should address this observation.

Response: thanks so much for this suggestion. We think it is mainly because the degree of staining for the enamel and the roots are different. The color on the enamel is obviously slighter than that on the roots, so after vibration for the same time, it is easy for the enamel to be whitened. In order to solve this problem, we prolonged the time of vibration. After vibration for 10 h, both the enamel and the roots became white obviously. The results can be seen in Figure 5 and Supplementary Figure 6-8 and also be described in page 16:

“It is worth noting that the color of roots is much deeper than enamel because of a higher concentration of staining agents on the roots [51]. It can be seen that the color of roots turns lighter but the stains still remain with vibration 3 hours, while the whole tooth can be completely whitened in the BTO nanoparticle turbid liquid with vibration 10 hours. The time-dependent whitening effect evidences the working principle of piezo-catalysis for tooth whitening.”

We also correct the vibration time in method, in page 23, “After stirring for 30 min, the beakers were ultrasonically vibrated for 10 hours”

Question 4: There are spectrometric methods available that use the CIELab system to observe enamel color changes and quantify color change. Use of quantification systems would help in describing the effects of the whitening reactions.

Response: thank you for your valuable comment. This is important to quantify the whitening results, so we take your advice to use the VITA Easyshade CIELab system to observe enamel and quantify color change. Furthermore, we calibrated the photos using a standard grayscale card. For this test, the method has been added in page 22: “The chroma of tooth enamel was periodically sampled by computer-aided shade matching (VITA Easyshade).”

In page 13, Figure 4 a-e showed the quantified whitening results have been added in page 14.

“In order to quantitatively characterize the whitening effect, Commission Internationale De L’Eclairage (CIELab) system was employed to quantify the color change. This system uses three variables: i.e., luminance L represents the difference between light ($L = 100$) and dark ($L = 0$); a and b designate the color values on the red-green axis and blue-yellow axis, respectively⁵⁰. The chroma of the tooth at each vibration condition and various vibration time was characterized, and the values of L , a , b are given in Fig. 4 b-d. It is obvious that the value of L increased, and the values of a and b decreased with vibration time for the BTO nanoparticle turbid liquid condition, while they show much weaker change for the deionized water condition, indicating the teeth has been obviously whitened by the BTO nanoparticle-based

piezo-catalysis effect. The color difference of ΔE was calculated by Eq.5 to further verify the whitening effect. As shown in Fig. 4e, the value of ΔE for both vibration condition increased with vibration time, but is roughly 3 times larger for the BTO nanoparticle turbid liquid than that for the deionized water. Thus, the quantitative CIELab measurements agree well with the color contrast in Fig. 4a.

$$\Delta E = \sqrt{\Delta L^2 + \Delta a^2 + \Delta b^2} \quad (5)$$

Question 5: Does the enamel surface become softer (Knoop Hardness) or change in any way?

Response: thank you for this comment. We checked the hardness of enamel using a Vickers microhardness tester, and the details can be found in supporting information Supplementary Fig. 12. The results reveal that there is no hardness change of the tooth enamel. The results shown in Supplementary Figure 12.

Supplementary Fig. 12 Vickers microhardness test using three different samples, **a** sample 1, **b** sample 2, **c** sample 3. The inset shows the position we test the hardness and the typical Vickers indentation we got. The Vickers hardness of enamel shows no change in all samples. Scale bars are 1 cm.

“Three teeth were randomly selected. Each tooth was cut into two pieces along the buccolingual division line using a line cutting machine. The teeth were then mounted in self-setting polymeric resin at room temperature. The samples were ground into a flat enamel surface successively using 600, 1000, 1200 grit silicon carbide paper, then polished with 0.2 and 0.05 microns alumina slurry. In order to avoid local overheating

on enamel surface, both grinding and polishing procedures were performed in water. Hardness was measured prior to staining, after staining and after whitening using a Vickers microhardness tester (Shanghai Huiju, HVS-1000Z) under a load of 200 g with 10 s dwell time. As shown in Supplementary Fig. 12, we choose five points on the enamel at different position, the hardness of tooth is about 300HV, and the value at different position is similar, which means the tooth is complete and healthy. Most importantly, the hardness shows no change at each step during the whole test, this proved no change in microstructure of the tooth.”

This part was discussed in page 18.

“To further explore the non-destructive nature piezo-catalysis, Vickers microhardness was measured during piezo-catalysis based tooth whitening. Three teeth were chosen randomly, and the hardness of enamel was measured under origin, stained and whitened states as shown in Supplementary Fig. 12. The hardness is about 300HV for all the teeth, and the value is stable across the surface of the tooth. It is obviously that the hardness of all the teeth has no variation during both staining and whitening process. The stable structural hardness also reveals that tooth whitening results from the piezo-catalysis effect.”

For future studies:

- Does the whitening occur in a similar fashion for teeth that have not been dehydrated?
- Does the whitening occur in a similar fashion for teeth stained with other chromogens? I use a combination of black tea, blueberry juice, red wine and fruit punch (Gatorade) and soak the teeth for a week.
- I have seen that dental adhesives and sealants are slowly degraded by peroxide bleaching - does the BTO system alter these polymers?
- I have also seen dental adhesives and sealants take up stains – can the BTO system be incorporated into these materials to prevent stains?

Response: Thank you for your advices for our future studies. We will consider your

suggestions with our co-authors (i.e., Prof. Xuehui Zhang and Dr. Yunyang Bai) and continue this work, who are working in Key Laboratory for Dental Materials & Dental Medical Devices.

Reviewers' Comments:

Reviewer #1:

Remarks to the Author:

I am satisfied with the author's answer and would be happy to recommend the publication of this manuscript.

Reviewer #2:

Remarks to the Author:

In this manuscript, the piezo-catalysis is proposed for tooth whitening. After revision, the manuscript is largely improved. However, there are still some concerns.

(1)The piezo-catalysis effect over tetragonal BTO nanoparticles is well documented for organic removal, bacterial inactivation, water splitting, etc., so the reviewer don't think it is still necessary to consider the piezo-catalysis effect by RhB removal experiments. The piezo-catalysis assisted decomposition of the same chemicals for forming tooth stains (such as the possible chemicals showing in Fig 4) would be more informative.

(2)As mentioned, the sources and chemical compositions leading to the staining of the teeth is complex, the involved chemical reactions would be complex during piezo-catalysis assisted tooth whitening. The piezo-catalysis assisted tooth whitening mechanism and safety evaluation need more investigations.

(a) It was supposed that some reactive species(ROS), such as $\bullet\text{OH}$ and $\bullet\text{O}_2^-$, would be generated during piezo-catalysis using BTO as catalyst, but their generation and concentration was not detected.

(b)The chemical compositions of dark stains are complex, and the ROS produced by piezo-catalysis is unknow (possibly more complicated than that by H_2O_2 treatment), therefore, the reaction intermediate during piezo-catalysis reaction will be complex, and their potential toxicity is worrisome, not to mention the possible leakage of Ba^{2+} during long-term strong vibration.

(c) As compared to the treatment by 30% (Fig 5) or 15% (Fig 6) H_2O_2 mentioned in the manuscript, the suitable concentration of generated reactive oxidative species (ROS) would be crucial for effective tooth whitening while avoiding tooth injury. But, how to make sure the concentration of ROS is behind the safety line?

(d) It is mentioned in the manuscript that (page 15) "tooth whitening effect under constant vibration was more significant than that under discontinuous vibration even for the same time", and that (page 16) "The whitening effect realized using a tooth brush is likely not as strong as that of ultrasonic vibration", probably because of less BTO and/or less ROS production. However, normally, the daily toothbrushing is just lasting for every 30 min or less by discontinuous vibration mode. In this regard, it is believed that the tooth whitening by long term discontinuous toothbrushing vibration will not as satisfying.

(e)Moreover, the experiment design for Fig 5 fail to consider the possible tooth destruction upon long-term accumulation effect.

(3)How about the stability of the BTO based piezo-catalysis system during tooth whitening,especially for those poled BTO?

Reviewer #3:

Remarks to the Author:

I am impressed with the improvements that the authors have made within the manuscript. All of my points/concerns were addressed satisfactorily.

The only missing piece is that there is no description of the statistical analyses that were done. I suggest to include "(NS = $p > 0.05$; ANOVA)" in the caption for Supplementary Figure 12.

Similarly mark the significant difference between vibration and BTO vibration with an astrisk in Figure 4 "(* = $p < 0.0X$; t-test at each timepoint)", and also a similar statement in the caption of Figure 6 should be included.

Clifton Carey

Dear Dr. Robert Guillatt and referees,

We are truly grateful for your high evaluation of our revised manuscript and the critical comments and thoughtful suggestions. Based on these comments and suggestions, we have made careful modifications on the 1st revised manuscript. All changes made to the text are in red color. We hope the 2nd manuscript will meet your magazine's standard. Below you will find our point-by-point responses to the reviewers' comments/questions below:

Reviewer #1:

I am satisfied with the author's answer and would be happy to recommend the publication of this manuscript.

Response: thanks a lot for your high evaluation to our revised manuscript and sincerely appreciate for your time and effort in reviewing our manuscript.

Reviewer #2:

In this manuscript, the piezo-catalysis is proposed for tooth whitening. After revision, the manuscript is largely improved. However, there are still some concerns.

- (1) The piezo-catalysis effect over tetragonal BTO nanoparticles is well documented for organic removal, bacterial inactivation, water splitting, etc., so the reviewer don't think it is still necessary to consider the piezo-catalysis effect by RhB removal experiments. The piezo-catalysis assisted decomposition of the same chemicals for forming tooth stains (such as the possible chemicals showing in Fig 4) would be more informative.
- (2) As mentioned, the sources and chemical compositions leading to the staining of the teeth is complex, the involved chemical reactions would be complex during piezo-catalysis assisted tooth whitening. The piezo-catalysis assisted tooth whitening mechanism and safety evaluation need more investigations.
 - (a) It was supposed that some reactive species (ROS), such as $\bullet\text{OH}$ and $\bullet\text{O}_2^-$, would be generated during piezo-catalysis using BTO as catalyst, but their generation and

concentration was not detected.

- (b) The chemical compositions of dark stains are complex, and the ROS produced by piezo-catalysis is unknown (possibly more complicated than that by H₂O₂ treatment), therefore, the reaction intermediate during piezo-catalysis reaction will be complex, and their potential toxicity is worrisome, not to mention the possible leakage of Ba²⁺ during long-term strong vibration.
 - (c) As compared to the treatment by 30% (Fig 5) or 15% (Fig 6) H₂O₂ mentioned in the manuscript, the suitable concentration of generated reactive oxidative species (ROS) would be crucial for effective tooth whitening while avoiding tooth injury. But, how to make sure the concentration of ROS is behind the safety line?
 - (d) It is mentioned in the manuscript that (page 15) “tooth whitening effect under constant vibration was more significant than that under discontinuous vibration even for the same time”, and that (page 16) “The whitening effect realized using a tooth brush is likely not as strong as that of ultrasonic vibration”, probably because of less BTO and/or less ROS production. However, normally, the daily toothbrushing is just lasting for every 30 min or less by discontinuous vibration mode. In this regard, it is believed that the tooth whitening by long term discontinuous toothbrushing vibration will not as satisfying.
 - (e) Moreover, the experiment design for Fig 5 fail to consider the possible tooth destruction upon long-term accumulation effect.
- (3) How about the stability of the BTO based piezo-catalysis system during tooth whitening, especially for those poled BTO?

Comment 1: The piezo-catalysis effect over tetragonal BTO nanoparticles is well documented for organic removal, bacterial inactivation, water splitting, etc., so the reviewer don't think it is still necessary to consider the piezo-catalysis effect by RhB removal experiments. The piezo-catalysis assisted decomposition of the same chemicals for forming tooth stains (such as the possible chemicals showing in Fig 4) would be more informative.

Response: thank you for your thoughtful comments. We cannot agree with the reviewer's comment any more, and we put the degradation of RhB in the supporting information section to demonstrate that the piezo-catalysis is dependent on the piezoelectricity of the nanoparticles. In the 2nd revised manuscript, Indigo Carmine, a widely used food pigment was used, because this organic dye is a common food colorant in juice drinks, carbonated drinks, confectioned wine and candies, and it is compound known to form tooth stains.

The results can be found in Figure 3 of page 9, and more discussion is in Page 10 to page 12.

Fig. 3 Degradation properties of piezo-catalysis. UV-Vis absorption spectra of Indigo Carmine solutions at various vibration time for the **a** poled and **b** unpoled BTO nanoparticles. The inset in each Figure is a series of photographs of piezo-catalyzed

Indigo Carmine dye solution time progression from left to right. Piezo-catalytic degradation efficiency performance of the poled and unpoled BTO nanoparticles in **c** direct concentration ratio C/C_0 and **d** logarithmic relationship of $\ln(C_0/C)$ by fitting with a linear function. **e** Electron paramagnetic resonance spectra (EPR) of radical spin-trapped by 5,5-dimethyl-1-pyrroline N-oxide (DMPO) over different piezo-catalysts in aqueous dispersion (top) and dimethyl sulfoxide (DMSO) dispersion (bottom).

“To demonstrate that the piezo-catalysis effect of BTO nanoparticles can be employed for tooth whitening, the degradation of Indigo Carmine solution was investigated using BTO turbid liquid with a concentration of 1 mg mL⁻¹ under ultrasonic vibration. Indigo Carmine was selected as a representative dye since it is a common food colorant used in juice drinks, carbonated beverages, confectionery wine and candies, and thus plays a key role in tooth staining⁴⁷.”

“As can be seen in Fig. 3c, more than 90% Indigo Carmine was degraded after ultrasonic vibration for 35 min by the poled BTO nano-catalysts.”

“It was found that the degradation rate of poled BTO ($k = 0.059 \text{ min}^{-1}$) is about 30 times as that of the unpoled BTO ($k = 0.002 \text{ min}^{-1}$).”

“As Rhodamine B (RhB) has been widely used to identify the catalysis effect, the degradation of RhB was also carried out using poled and unpoled BTO, respectively. Results similar to the Indigo Carmine test were observed (Supplementary Fig. 4). The degradation rate of RhB using poled BTO ($k = 0.448 \text{ h}^{-1}$) is about 8 times higher than that of the unpoled BTO ($k = 0.062 \text{ h}^{-1}$).”

Comment 2(a): It was supposed that some reactive species (ROS), such as $\bullet\text{OH}$ and $\bullet\text{O}_2^-$, would be generated during piezo-catalysis using BTO as catalyst, but their generation and concentration was not detected.

Response: thanks a lot for this valuable suggestion. We detected the reactive species (ROS) by electron paramagnetic resonance (EPR). The result shows that both $\bullet\text{OH}$ and $\bullet\text{O}_2^-$ have been detected during the piezo-catalysis using poled BTO as catalyst. The DMPO spin-trapping ESR spectra of poled BTO has been added in page 9, Figure 3.

Fig. 3 Degradation properties of piezo-catalysis. UV-Vis absorption spectra of Indigo Carmine solutions at various vibration time for the **a** poled and **b** unpoled BTO nanoparticles. The inset in each Figure is a series of photographs of piezo-catalyzed Indigo Carmine dye solution time progression from left to right. Piezo-catalytic degradation efficiency performance of the poled and unpoled BTO nanoparticles in **c** direct concentration ratio C/C_0 and **d** logarithmic relationship of $\ln(C_0/C)$ by fitting with a linear function. **e** Electron paramagnetic resonance spectra (EPR) of radical spin-trapped by 5,5-dimethyl-1-pyrroline N-oxide (DMPO) over different piezo-catalysts in aqueous dispersion (top) and dimethyl sulfoxide (DMSO) dispersion (bottom).

The results of the reactive specie detection were added in page 13.

“As the reactive species of $\bullet\text{OH}$ and $\bullet\text{O}_2^-$ are necessary for piezo-catalysis effect, the

generation of $\bullet\text{OH}$ and $\bullet\text{O}_2^-$ was verified by the electron paramagnetic resonance (EPR) technique using 5,5-dimethyl-1-pyrroline N-oxide (DMPO) as spin trapper. The signature peaks of both DMPO- $\bullet\text{OH}$ (top of Fig. 3e) and DMPO- $\bullet\text{O}_2^-$ (bottom of Fig. 3e) were detected, and the intensities of the EPR signals increased with vibration time. In addition, the EPR signals are dependent on piezoelectricity (i.e., poled PMN-PT > poled BTO > unpoled BTO > cubic BTO), which is consistent with the results of RhB degradation experiment (Supplementary Fig. 4 and 6).”

Method of ROS detection was added in page 23.

“**Detection of reactive species.** The reactive species created by the piezocatalyst were detected by the electron paramagnetic resonance (EPR) technique with a Bruker EMX-10/12 spectrometer. First, 10 mg samples were dissolved in 10 mL of deionized water or 10 mL of dimethyl sulfoxide for $\bullet\text{OH}$ and $\bullet\text{O}_2^-$ detection, respectively. After 15 min of vibration, 200 μL solution was taken out and 20 μL of 5,5-dimethyl-1-pyrroline N-oxide (DMPO) was added into the solution. The reactive species were detected immediately after ultrasound for 0, 1 and 5 min. For the $\bullet\text{OH}$ in 3% H_2O_2 , we used Fenton reaction to release the reactive species. Briefly, 10 μL of 5 mmol L^{-1} FeSO_4 was mixed with 20 μL DMPO, and 180 μL of 3% H_2O_2 was added in the mixed solution. Under ultrasonic vibration for 30 s, the $\bullet\text{OH}$ created by this classical reaction was detected.

”

Comment 2(b): The chemical compositions of dark stains are complex, and the ROS produced by piezo-catalysis is unknown (possibly more complicated than that by H_2O_2 treatment), therefore, the reaction intermediate during piezo-catalysis reaction will be complex, and their potential toxicity is worrisome, not to mention the possible leakage of Ba^{2+} during long-term strong vibration.

Response: thanks a lot for this valuable comment. We have identified the ROS by EPR, the EPR spectra has been added in page 9, Figure 3, and the related discussion was added in page 13, i.e., “As the reactive species of $\bullet\text{OH}$ and $\bullet\text{O}_2^-$ are necessary for piezo-

catalysis effect, the generation of $\bullet\text{OH}$ and $\bullet\text{O}_2^-$ was verified by the electron paramagnetic resonance (EPR) technique using 5,5-dimethyl-1-pyrroline N-oxide (DMPO) as spin trapper. The signature peaks of both DMPO- $\bullet\text{OH}$ (top of Fig. 3e) and DMPO- $\bullet\text{O}_2^-$ (bottom of Fig. 3e) were detected, and the intensities of the EPR signals increased with vibration time. In addition, the EPR signals are dependent on piezoelectricity (i.e., poled PMN-PT > poled BTO > unpoled BTO > cubic BTO), which is consistent with the results of RhB degradation experiment (Supplementary Fig. 4 and 6).”

It can be seen that both $\bullet\text{OH}$ and $\bullet\text{O}_2^-$ has been detected as we discussed in the 2nd revised manuscript. The result and discussion of this part was added in page 13.

The reaction intermediate of Indigo Carmine has been widely studied for photocatalysis. Due to the reactive species created by piezo-catalysis is the same with that by photocatalysis, the process of Indigo Carmine degradation via piezo-catalysis can be conjectured to the same as photocatalysis. The degradation pathways of Indigo Carmine have been described in detail [*Materials Science in Semiconductor Processing* 2020, 105.]. *Indigo Carmine degradation was analyzed by analytical techniques, UPLC-PDA and HR-QTOF ESI/MS that helps to identify the degradation products, organic reactions (Hydroxylation, oxidation, methylation, decarboxylation, and desulfonation), and four pathways of Indigo Carmine in water. All the reaction intermediates were not harmful to human body.*

Fig. 9 Proposed degradation pathways of IC in water by visible light by Ni-BaMo₃O₁₀ photocatalyst.

[Materials Science in Semiconductor Processing 2020, 105.]

To explore the possibility of Ba²⁺ leakage during vibration, Ba levels were explored after 30 minutes of continual vibration (supplementary Fig. 17). Since a typical recommendation for daily toothbrushing is only two minutes, and this is performed using manual vibration, 30 minutes of continuous ultrasonic vibration far exceeds typical vibration levels. Since there is no detectable leakage of Ba²⁺ during continuous, high-energy excitation, we can conclude that there is little risk of such leakage during lower-energy manual vibration, even over the course of prolonged use.

The results were added as Supplementary Fig. 17. The related discussion was also added in page 21 of 2nd revised manuscript.

“Additionally, the possibility of Ba²⁺ leakage during vibration was also examined, and the result shows no detectable Ba²⁺ was created after vibrated for 30 min, which far exceeds typical daily toothbrushing vibration levels (Supplementary Fig. 17).”

Supplementary Fig. 17 **a** Schematic diagram of experimental process, **b** (left column) the concentration of total Ba element (including Ba^{2+} and compound) in the supernatant obtained from BTO (top) and BaSO_4 (bottom) before (red) and after (blue) vibration for 30 min; (middle column) the Ba element concentration difference of before-vibration and after-vibration for 30 min; (right column) concentration of total Ba element and the difference with experimental errors. **c** the SEM image shows particles with different size of BTO (top) and BaSO_4 (bottom). Scale bars in **c** are 200 nm (top) and 500 nm (bottom).

“In order to check the possible leakage of Ba^{2+} during vibration, 50 mg BTO nanoparticles were dispersed in 50 ml deionized water, and the obtained solutions were stirred for 30 min to make sure that the BTO nanoparticles were evenly dispersed. Then, the suspension was vibrated for 30 min (i.e., daily toothbrush time is less than 30 min). 10 ml suspension was collected at a certain time and centrifuged in order to remove the

nanoparticles. The supernatant was collected and this process repeated for three times in order to avoid experimental error. The concentration of barium in the supernatant was measured by Inductively Coupled Plasma-Optical Emission Spectroscopy (ICP-OES), which is used to measure the concentration of an element, including the existence of related ions and compounds. The barium meal (BaSO_4) is often used as gastrointestinal radiological examinations, indicating that BaSO_4 is safe, so the same experiment was taken using BaSO_4 as comparison. The left column of Supplementary Fig. 17 c shows that the barium element was detected in the supernatant of both BTO and BaSO_4 before and after vibration, while the concentration difference of barium element before-and after-vibration for both BTO and BaSO_4 supernatant changed randomly (middle column of Supplementary Fig. 17 c). Here, the concentration difference before and after vibration can be regarded as concentration of leakage Ba^{2+} . The obtained concentration of Ba element and the difference with experimental errors of before-vibration and after-vibration was given in the right column of Supplementary Fig. 17 c. The existence of Ba element in the supernatant is possible due to the incomplete removal of nanoparticles by centrifugation, because the concentration of barium element in supernatant of BaSO_4 was a little lower than that of BTO, which can be understood by particle size of BaSO_4 and BTO (i.e., ~ 300 nm relative to ~ 150 nm). These lead to less BaSO_4 nanoparticles remains in the supernatant after centrifugation. The comparison of the concentration difference of barium before-and after- vibration between BTO and BaSO_4 shows that the barium concentration difference is near zero and experimental error varies within the same range. In turn, we can conclude that the variation in barium concentration before and after vibration was caused by experimental errors rather than the leakage of Ba^{2+} .”

Actually, piezoelectric materials have been widely studied for biomedical applications. *The ability of energy generation with no external source, no electrode requirement, high sensitivity, low energy consumption, and the possibility for miniaturization make piezoceramics an interesting candidate for biotechnology. Some examples include insulin pumps and piezoelectric energy generators for implantable*

medical devices. Ferroelectrics have also the added advantage that apart from piezoelectricity, they feature the possibility to tailor surface charges through poling. This entails tunable topo-physical features such as surface energy and wettability. This opens a quite promising and complex multidisciplinary research field for lead-free ferroelectrics. Most studies on biocompatible piezoelectrics have focused so far on BT-based materials [Applied Physics Reviews 2017, 4 (4)].

The biocompatibility of BTO was claimed in several studies *in vitro* and *in vivo*. The *in vitro* studies have been done using mouse myoblast cells [J Mater Sci Mater Med 2015, 26 (2), 103.], mouse osteoblast cells [J Biomed Mater Res A 2014, 102 (7), 2089-95.] and rat mesenchymal stem cells [Colloids Surf B Biointerfaces 2013, 102, 312-20.]. The results show that BTO nanoparticles have no effect on cell growth, proliferation and differentiation.

Fig. 3 Representative fluorescent microscopy images of the cultured mouse myoblast cells, exposed with the different concentrations of HA-40 wt % BaTiO₃ eluates: a control, b 0.25 mg/ml, c 2.5 mg/ml and d 25 mg/ml, at 24 h of post-exposure period; e control, f 0.25 mg/ml, g 2.5 mg/ml and h 25 mg/ml, at 48 h of post-exposure period

The study of BTO biocompatibility using mouse myoblast cells. [J Mater Sci Mater Med 2015, 26 (2), 103.]

FIGURE 6. LDH levels of 7F2 cells cultured directly on BT and bioglass foams after 72 h. These cytotoxicity results show that BT foams are significantly ($p < 0.05$) less toxic than cells cultured with 1% Triton X-100 for 45 min. Additionally, there is no significant difference between the cytotoxicity of BaTiO₃ and 45S5 Bioglass. Values presented are averages \pm standard deviation, $n = 4$. Cross-hatched area represents the baseline toxicity of cells that were left untreated in nonsupplemented complete media. [Color figure can be viewed in the online issue, which is available at wileyonlinelibrary.com.]

The study of BTO biocompatibility using mouse osteoblast cells [*J Biomed Mater Res A* 2014, 102 (7), 2089-95.]

Fig. 3. Live/dead assay, early apoptosis detection, and ROS detection performed after 120 h of incubation of MSC cultures with increasing concentrations of glycol-chitosan coated BTNPs; no evidence of appreciable negative effects are detectable in each test.

The study of BTO biocompatibility using rat mesenchymal stem cells. [*Colloids Surf B Biointerfaces* 2013, 102, 312-20.]

The *in vivo* test was taken on mice using hydroxyapatite (HA)-BaTiO₃ eluate, three groups of mice were intra-articularly injected with 100 μ L of eluate at concentrations of 0.25, 2.5 and 25 mg/ml, respectively. *It is the first result to conclusively confirm the non-toxic effect of HA-BaTiO₃ piezobiocomposite nanoparticulates, in vivo [J Mater Sci Mater Med 2015, 26 (2), 103.]*

Fig. 4 Representative histological images of the mice heart sections, at 7 days post-injection with HA-40 wt % BaTiO₃ particle eluates, showing normal appearance of cardiac muscle in both, a control and b C3 (25 mg/ml) groups

Fig. 5 Histological images of mice liver tissue, at 7 days post-injection with HA-40 wt % BaTiO₃ particle eluates, showing intact and normal architecture of hepatocytes, sinusoids and central vein in both, a control and b C3 (25 mg/ml) groups

Fig. 6 Histological sections of mice kidney, at 7 days post-exposure to HA-40 wt % BaTiO₃ particle eluates, showing normal appearance of renal cortex with glomerular tufts and tubules in both, a C1 (0.25 mg/ml) and b C3 (25 mg/ml) groups

Fig. 7 Photomicrograph revealing the histological features of mice spleen sections, at 7 days post-exposure to HA-40 wt % BaTiO₃ particle eluates, showing normal appearance of splenic pulp vein in both, a control and b C3 (25 mg/ml) groups

Fig. 8 Histopathological features of mice knee joint sections, at 7 days post-injection with HA-40 wt % BaTiO₃ particle eluates: a and b show regions of particle accumulation between the cartilage and skeletal muscle [C1 group (0.25 mg/ml)], c and d depict higher amount of particle accumulation in the fibroadipose tissue region in

the vicinity of knee joint [C3 group (25 mg/ml)]. Black arrows indicate intact cartilage without any interruption by the injected particles, blue arrows provide clear indication of particles phagocytosed by the macrophages close to the articular cartilage (Color figure online)

The histopathological images above are form the refs. [J Mater Sci Mater Med 2015, 26 (2), 103.]

Histological images of major organs of mice, at 7 days post injection with HA-40 wt % BaTiO₃ particle eluates. *The in vivo biocompatibility assessment confirmed that the injected particles were not translocated to any of the major organs such as liver, kidney, lung, spleen, heart in the treated mice. The histopathological analysis did not reveal any alteration in the tissue structures of the vital organs of experimental mice.*

Additionally, *the entire injected volume of HA-40 wt % BaTiO₃ eluates was found to be accumulated near the knee joint, encircled with densely arranged macrophages that have partially phagocytosed the particles. There was no notable formation of fibrous tissue around the particles and the eluates remained in direct contact to the tissues, without any observable recruitment of foreign body giant cells and mast cells. More importantly, the injected particles have not induced any foreign body/inflammatory reactions. [J Mater Sci Mater Med 2015, 26 (2), 103].*

The study of the acute toxicity of BTO administered orally was made on the mouse and on the rat. *In the rat studies no signs of toxicity were observed. It can therefore be concluded that barium titanate are substantially non-toxic substances, and at the dosage levels used for radiological examination of the digestive tract according to the instant invention are totally lacking in toxicity [US Patent 4,020,152, 1977].*

Number of mice	Barium titanate g/kg (orally administered)	Mortality after		
		4 hrs.	24 hrs.	72 hrs.
3	1	0	0	0
3	2	0	0	0
3	4	0	0	0
10	8	0	0	0
10	12	0	0	1
10	16	0	2	5

The result of the acute toxicity of BTO administered orally was from ref. *[US Patent 4,020,152, 1977].*

As the remains of BTO in our experiment were only ~4 mg/L, the concentration is much lower than that used in the toxicity test *in vivo*. Considering that the BTO suspension used in each simulated tooth brushing was only 50ml, the theoretical residue was only 0.2mg which is far less than used in oral tests. It is worth noting that the

amount of BTO that actually goes into the mouth and is taken orally into the body is even less. In this way, The BTO nanoparticles is safe enough for tooth whitening.

Comment 2(c): As compared to the treatment by 30% (Fig 5) or 15% (Fig 6) H₂O₂ mentioned in the manuscript, the suitable concentration of generated reactive oxidative species (ROS) would be crucial for effective tooth whitening while avoiding tooth injury. But, how to make sure the concentration of ROS is behind the safety line?

Response: thank you for this valuable comment. We have discussed with dentist about the ROS safety line and check it from various dentistry standard (such as international standard ISO 28399, as shown below), but there is no exact safety line for the ROS.

**INTERNATIONAL
STANDARD**

**ISO
28399**

First edition
2011-01-15

Dentistry — Products for external tooth bleaching

Médecine bucco-dentaire — Produits d'éclaircissement dentaire, à usage externe

However, it is well known that 3% hydrogen peroxide is widely used in oral cleaning and wound treatment, so it can be believed that the ROS produced by 3% is safe to oral

soft tissue. Here we performed the tooth whitening using 3% H₂O₂ and poled BTO as whitening agent for 10 hours, respectively. The tooth enamel shows no change in BTO suspension, while the tooth enamel was damaged by 3% H₂O₂. The result has been added in Supplementary Fig. 14. The results of this experiment were also discussed in page 19.

Supplementary Fig. 14 Nondestructive characterization. The scanning electron microscope images of an identical tooth **a** before whitening treatment, **b** after piezo-catalysis whitening in BTO turbid liquid for 10 hours, after further whitening by **c** 3% H₂O₂ for 10 hours and **d** 30% H₂O₂ for 2 hours. The images at the bottom are the enlarged view of the marked identical zone. Scale bars are 100 μm (top) and 50 μm (bottom).

“A similar experiment was also carried out using a 3% hydrogen peroxide solution, which is often used clinically to clean the mouth. SEM images show that the degree of enamel damage in the 3% hydrogen peroxide solution is noticeably progressed compared with treatment using BTO (see Supplementary Fig. 14). In the 3% hydrogen peroxide solution, the tooth enamel was eroded and extensive damaged, exhibiting pitting and holes. The relative lack of enamel deterioration using BTO nanoparticles further indicates that the reactive species created by piezo-catalysis of BTO nanoparticles are less than that created by 3% H₂O₂ during the tooth whitening process.”

We also tested the concentration of 3% H₂O₂ by EPR. Fenton reaction was used to release the •OH, and the EPR spectrum of DMPO-•OH was added in Supplementary Fig. 15.

Supplementary Fig. 15 The EPR spectrum of DMPO-•OH created by Fenton reaction in 3% H₂O₂ (top) and poled BTO after vibrated for 5 min (bottom).

“It is well known that 3% H₂O₂ is widely used medically to clean mouth, here the concentration of reactive species in 3% H₂O₂ was used as a benchmark of safety. In order to identify the safety of BTO for tooth whitening, we compared the concentration of •OH created by BTO after vibration for 5 min and that created by Fenton reaction in 3% H₂O₂. The concentration of •OH in 3% H₂O₂ is about 50 times larger than that created by BTO after vibration for 5 min. This further evidences that the BTO nanoparticles used for tooth whitening have no risk of excessive free radicals.”

Comment 2(d): It is mentioned in the manuscript that (page 15) “tooth whitening effect under constant vibration was more significant than that under discontinuous vibration even for the same time”, and that (page 16) “The whitening effect realized using a tooth brush is likely not as strong as that of ultrasonic vibration”, probably because of less BTO and/or less ROS production. However, normally, the daily toothbrushing is just lasting for every 30 min or less by discontinuous vibration mode. In this regard, it is believed that the tooth whitening by long term discontinuous toothbrushing vibration will not as satisfying.

Response: thank you for your valuable comment. Actually, experiment in Figure 4 is realized in discontinuous vibration mode based on an electric toothbrush. It is important to note that all the electric toothbrushes have safety mode to avoid the damage on oral tissue arising from prolonged vibration. Electric toothbrushes, therefore, cannot create continuous vibration. Technically speaking, the vibration we used in the tooth whitening experiment was discontinuous. The tooth brushed using BTO turbid liquid shows a whitening effect, while in contrast, there was no perceptible change in the color of the one brushed in deionized water alone. The result proved that the tooth can be whitened by BTO nanoparticles under the condition of long-term use, even under the discontinuous vibration. In page 17, we added this part to explain the test in detail.

“To more realistically simulate daily toothbrushing, the tooth was vibrating for periodically at 2 minute intervals for 10 hours.”

Comment 2(e): Moreover, the experiment design for Fig 5 fail to consider the possible tooth destruction upon long-term accumulation effect.

Response: thank you for your valuable comment. In order to make sure that the long-term accumulation effect shows no damage to the tooth enamel, we extend the vibration time to 10 hours for poled BTO nanoparticles. Please note that the stained tooth can be notably whitened after 3 hours and completely whitened after 10 hours (Fig. 4a). The SEM images based on different whitening agents were shown in Supplementary Fig. 14. The related discussion was added in page 19.

Supplementary Fig. 14 Nondestructive characterization. The scanning electron microscope images of an identical tooth **a** before whitening treatment, **b** after piezo-catalysis whitening in BTO turbid liquid for 10 hours, after further whitening by **c** 3% H₂O₂ for 10 hours and **d** 30% H₂O₂ for 2 hours. The images at the bottom are the enlarged view of the marked identical zone. Scale bars are 100 μm (top) and 50 μm (bottom).

In page 19, we added the discussion as:

“while another group of SEM images under vibration for 10 hours was presented in Supplementary Fig. 14. Please note that the stained tooth can be notably whitened in 3 hours and completely whitened within 10 hours (Fig. 4a). The SEM image of microscopic morphology shows that the piezo-catalytic tooth whitening is mechanically nondestructive to the tooth enamel.”

Comment 3: How about the stability of the BTO based piezo-catalysis system during tooth whitening, especially for those poled BTO?

Response: thank you for your valuable comment. We tested the stability of poled BTO catalyst. We did the RhB degradation test and tooth whitening test three times using the same BTO. The results showed a great stability for BTO piezo-catalyst, the data of BTO stability for RhB degradation has been added in Supplementary Figure 4. This part has been added in page 12 in manuscript.

“Furthermore, the recyclability of poled BTO for degrading RhB under an ultrasonic vibration was investigated. It was found that the degradation efficiency exhibits no obvious changes after undergoing three recycling processes, indicating the stability of piezo-catalysts for long-term use.”

Supplementary Fig. 4 Degradation properties of piezo-catalysis. UV-Vis absorption spectra of RhB solutions at various vibration time for the **a** poled and **b** unpoled BTO nanoparticles. The inset in each Figure is a series of photographs of piezo-catalyzed RhB dye solution progressing in time from left to right. Piezo-catalytic degradation efficiency performance of the poled and unpoled BTO nanoparticles in **c** direct concentration ratio C/C_0 and **d** logarithmic relationship of $\ln(C_0/C)$ by fitting with a linear function. **e** Recycling ability of poled BTO for degrading RhB under an ultrasonic vibration.

The stability of poled BTO for tooth whitening was also tested, as shown in

Supplementary Figure 11, the whitening effect shows negligible decrease. The result has been added in page 16 of main text.

“Besides the stability of organic dye degradation (Supplementary Fig. 4e) and structure of BTO, the continued ability of BTO nanoparticles after prolonged vibration time to whiten teeth was also characterized. Three different stained teeth were successively whitened by the same poled BTO under the same vibration time of 10 h, while the tooth whitening effect of the same poled BTO nanoparticle shows no obvious changes (Supplementary Fig. 11).”

Supplementary Fig. 11 The stability of poled BTO based piezo-catalysis system for tooth whitening. The tooth was vibrated for 10 h in BTO suspension. Scale bars are 1cm.

The structure stability of nano-sized BTO was also checked, the result shows in Supplementary Fig. 5.

Supplementary Fig. 15 XRD pattern of the BTO nanoparticles before and after several

times catalyst test. No additional peaks were observed after the piezo-catalysis process.

“The structural stability of nano-sized BTO also serves as evidence that the degradation of **organic dye** results from the piezo-catalysis effect of BTO, rather than any chemical reaction between BTO and **organic dye** (see XRD data shown in Supplementary Fig. 5).”

Reviewer #3:

I am impressed with the improvements that the authors have made within the manuscript. All of my points/concerns were addressed satisfactorily.

The only missing piece is that there is no description of the statistical analyses that were done. I suggest to include "(NS = $p > 0.05$; ANOVA)" in the caption for Supplementary Figure 12. Similarly mark the significant difference between vibration and BTO vibration with an asterisk in Figure 4 "($* = p < 0.05$; t-test at each timepoint)", and also a similar statement in the caption of Figure 6 should be included.

Response: thank you for the high evaluation to our revised manuscript. We have added the statistical analyses in the 2nd revised manuscript for the figure captions.

Fig. 4 Demonstration of tooth whitening based on piezo-catalysis effect. **a** Photographs of teeth under treatment of vibration in (top) pure deionized water and (bottom) turbid liquid of BTO nanoparticles for 0, 1, 3 and 10 hours, respectively. These photographs are successive images of the same tooth. Variation in **b** luminance L , **c** color value of red-green axis a , **d** color value of blue-yellow axis b and **e** color difference ΔE at vibration time of 0, 1, 3 and 10 hours. **f** The setup we used to simulate the daily teeth cleaning behaviors. **g** Photograph of the teeth brushed with pure deionized water (top) and BTO nanoparticles turbid liquid (bottom), respectively. The comparison of brushed zone, marked by circles, reveals that the piezo-catalysis with electric toothbrush was effective to tooth whitening. **The difference of ΔE at each time point was calculate by t-test, $p < 0.01$.** Scale bars are 1 cm.

Fig. 6 Cytotoxicity characterization. Cell proliferation and morphology with different tissue culture medium for 1, 2, 3 days. AO/EB staining of A7r5 cells grown in **a** pure tissue culture, **b** tissue culture with BTO nanoparticles in concentration of 1 mg mL^{-1} , and **c** tissue culture with H_2O_2 in volume fraction of 15%. **d** Viability of A7r5 cells in different tissue culture measured by the MTT assay. **NS = $p > 0.05$; *** = $p < 0.01$ (t-test at each timepoint)**. Scale bars are $200 \text{ }\mu\text{m}$.

The figure caption of Supplementary Figure 12 was also modified.

Supplementary Fig. 15 Vickers microhardness test using three different samples, **a** sample 1, **b** sample 2, **c** sample 3. The inset shows the position we test the hardness and the typical Vickers indentation we got. The Vickers hardness of enamel shows no change in all samples. **(NS = $p > 0.05$; ANOVA)**. Scale bars are 1 cm .

Reviewers' Comments:

Reviewer #2:

Remarks to the Author:

I am satisfied with the improvement of the revised manuscript.

Some minor errors can be further corrected, such as, page 4 line 11 "will dispersed", page 4 line 15 "are absorbed", page 5 Figure 1c "degradation production", page 6 line 5 "absorption"...Please be careful.